*Report*

**EMBO** *reports*

# PIF/harbinger transposon-derived protein promotes 7SL expression to enhance pathogen resistance

Shang Geng [1], Xing Lv[1] & Tianjun Xu [1,2,3]✉

## Abstract

Transcriptional regulation governs gene expression levels, primarily controlled by "cis-acting DNA elements" and "trans-acting protein factors". However, the conventional view that cis-regulation is solely attributable to DNA elements is challenged in this study. Our research indicates that transposon-derived proteins may retain their original DNA-binding preference and exert cis-regulatory effects on nearby genes on the chromosome, thus denoted as "cis-acting factors". Specifically, we show that the ADF-1L protein, derived from the PIF/harbinger transposon, recruits the histone acetyltransferase KAT2B in a MADF domain-dependent manner, facilitating its own nuclear translocation and binding to and cis-regulating its own and adjacent gene 7SL-23. ADF-1L protein also boosts the host's resistance to pathogens by promoting the expression of immune molecule 7SL RNA. In summary, our findings expand the types of molecules that can exert cis-function in gene regulation and underscore the relevance of transposons-derived sequences in cellular processes.

**Keywords** 7SL; PIF/harbinger; KAT2B; Cis-acting Factor; Immune
**Subject Categories** Chromatin, Transcription & Genomics; Microbiology, Virology & Host Pathogen Interaction

## Introduction

Transcriptional regulation is a critical process in biological systems, governed by cis-acting DNA elements and trans-acting protein factors. Nonetheless, the traditional perspective that attributes cis-regulation exclusively to DNA elements has been reevaluated, with recent studies emphasizing the important cis-regulatory functions of long noncoding RNAs (lncRNAs) (Kopp and Mendell, 2018). These lncRNAs can influence the expression and chromatin states of neighboring gene in chromosome through several mechanisms: by recruiting regulatory factors, through transcription and splicing processes independent of the RNA transcript, or via DNA elements within their promoter or gene locus (Anderson et al, 2016; da Rocha and Heard, 2017; Huarte et al, 2010; Latos et al, 2012).

Similar to cis-acting lncRNAs, the identification of proteins that can cis-regulate neighboring gene, termed "cis-acting factors", is critical for enhancing our understanding of the host's transcriptional regulation mechanism.

Transposable elements (TEs) are DNA fragments capable of replication or transposition, often composing a substantial portion of the eukaryotic genome. These TEs frequently harbor sequences adept at enlisting host transcription apparatus to express their own products and encourage transposition (Ichiyanagi and Okada, 2008). Over time, some TEs have lost the ability to replicate and transpose in the host genome due to mutations or truncations in the sequences required for their migration. Despite this loss, these "domesticated" TEs can still contribute to genomic innovation by dispersing cis-acting elements, such as transcription factor binding sites, which facilitate the emergence of new gene regulatory networks (Davidson and Britten, 1979; Dechaud et al, 2019; Sundaram and Wysocka, 2020). More recently, it has been recognized that the proteins encoded by TEs themselves provide additional avenues for host transcriptional regulation (Cosby et al, 2021; Feschotte, 2008; Fueyo et al, 2022). Paired box (PAX) proteins, which are crucial transcription factors involved in embryonic development, tissue differentiation, and organogenesis, feature DNA-binding domains believed to have evolved from transposases (Cosby et al, 2021; Feschotte, 2008; Fueyo et al, 2022). In addition, an examination of ~600 tetrapods genomes indicates that the merging of transposase domains with host regulatory domains is a recurring mechanism for the formation of new transcription factors during the process of evolution (Cosby et al, 2021). Due to the fact that the development of novel transcription factors frequently plays a crucial role in the emergence of significant evolutionary innovations, it is alluring to hypothesize that the acquisition of transposition proteins has had a profound influence on phenotypic diversification (Feschotte, 2008).

Histone acetylation modification plays a critical role in regulating chromatin structure and function in the transcriptional system, and it is commonly associated with transcriptional activation. The acetylation status of a specific chromatin locus is regulated by two opposing categories of histone-modifying enzymes, namely histone acetyltransferases (HATs) and deacetylases (HDACs). HATs add acetyl groups to target histones, while HDACs remove them. Among the HATs, human KAT2A (GCN5) and its paralogue KAT2B (PCAF) are extensively studied. These enzymes primarily acetylate histone H3, and to a lesser extent,

[1]Laboratory of Fish Molecular Immunology, College of Fisheries and Life Science, Shanghai Ocean University, Shanghai, China. [2]Laboratory for Marine Biology and Biotechnology, Qingdao Marine Science and Technology Center, Qingdao, China. [3]Marine Biomedical Science and Technology Innovation Platform of Lin-gang Special Area, Shanghai, China. ✉E-mail: tj-xu@shou.edu.cn

histone H4, resulting in changes in chromatin structure (Fournier et al, 2016). However, it should be noted that KAT2A/2B has the capability to acetylate non-histone substrates, including CDC6 and cyclin A, to modulate the G1/S cell cycle progression and mitosis (Orpinell et al, 2010; Paolinelli et al, 2009). Despite documented involvement in diverse cellular processes such as DNA replication, DNA repair, cell cycle progression, and cell death, a comprehensive compilation of the cellular targets of KAT2A/2B has not been completed (Sheikh and Akhtar, 2019).

The PIF/harbinger DNA TEs are widespread from plants to vertebrates but absent from mammalian genomes, full-length of their sequence encodes a transposase and another protein containing a predicted Myb/SANT domain that is necessary for cis-regulating its own transposon DNA sequence (Kapitonov and Jurka, 2004; Sinzelle et al, 2008). Our study suggested that ADF-1L, a protein derived from the Myb/SANT-domain-containing protein of PIF/harbinger, can serve as a "cis-acting factor" to bind and cis-regulate its own and adjacent gene 7SL-23, thereby countering the traditional notion that cis-regulation is exclusively driven by DNA elements. The concept of a "cis-acting factor" contrasts with transcription factors like P53 (Hudson et al, 1995) and OCT4/SOX2 (Schulz and Hoffmann, 2007), which regulate their own expression but do not affect neighboring genes. Their self-regulation seems to be a positive feedback loop preserved for its evolutionary advantage. In contrast, ADF-1L regulates both its own expression and that of adjacent genes, representing a unique form of cis-regulation. Mechanistically, ADF-1L recruits histone acetyltransferase KAT2B to the promoter region of the ADF-IL and 7SL-23 genes in a MADF domain-dependent manner, which led to enhanced transcription of the 7SL-23 gene. Furthermore, 7SL RNA, traditionally seen as a housekeeping gene, plays a role in immune regulation by modulating the miR-2187-3p/TRAF6 axis. ADF-1L enhances this process, improving host resistance to infections. Therefore, our findings expand the types of molecules that can exert cis-function in gene regulation and emphasize the important role of transposons in the cellular processes and the evolutionary trajectory of the biological genome.

## Results and discussion

### Upstream of the active 7SL lies a PIF/harbinger TE-derived ADF-1L gene

The 7SL gene, part of the short interspersed nuclear element (SINE) family of retrotransposons, can be transcribed by RNA polymerase III (Pol III) into approximately 300 nucleotides of RNA. While this gene typically exists in multiple copies within organisms, most have become inactive (Lunyak and Atallah, 2011). An analysis of 7SL gene duplicates across various vertebrates showed significant differences in copy numbers among species. Primate mammals, including humans, have over two thousand copies, far more than other animals, while the elephant shark, an ancient species, has only one 7SL gene (Fig. 1A). Additionally, fish that have undergone a fourth genome-wide replication, such as zebrafish and rainbow trout, have more 7SL copies than other fish, indicating alternative replication pathways besides retroposition (Fig. 1A). In this study, we used the miiuy croaker (*Miichthys miiuy*), a lower vertebrate with extensive transcriptome data and a chromosome-scale genome (Che et al, 2014; Geng et al, 2021; Geng et al, 2022a), to explore the

genetic regulatory mechanisms of 7SL. Genome scanning of *M. miiuy* revealed 24 7SL genes (Fig. 1B). By mapping RNA-seq data (Data ref: Geng et al, 2022b) to the genome, we identified a particularly active 7SL gene, 7SL-23, located on chromosome 19 (Fig. 1C). This gene exhibits the highest sequencing depth and shows significant upregulation following infection with *Siniperca chuatsi rhabdo* virus (SCRV) and *Vibrio anguillarum* (Fig. 1C). Sequence alignment revealed high evolutionary conservation between *M. miiuy* 7SL-23 and human 7SL, both retaining the conserved A and B boxes (Fig. 1D). Further analysis of 7SL expression in spleen tissues and MKC cells during SCRV or *V. anguillarum* infections supported the RNA-seq findings, showing significant upregulation of 7SL expression (Fig. 1E,F). Moreover, the *M. miiuy* 7SL, primarily cytoplasmic, exhibited decreased transcription with the Pol III inhibitor ML-60218, consistent with known 7SL characteristics (Fig. 1G,H). To investigate potential activation mechanisms for the active 7SL-23, we examined the genomic regions surrounding it and found a transcript approximately 800 bp upstream of 7SL-23 (Fig. 1I). This transcript encodes a protein with a MADF domain, which is a unique domain in the transcription factor ADF-1 protein of Drosophila (England et al, 1992; England et al, 1990); as such, we named this gene ADF-1L (Fig. 1I). This adjacency between Pol II gene ADF-1L and Pol III gene 7SL-23 mirrors a potential functional genetic arrangement observed in the genome of Trypanosoma brucei, where the Pol III genes hallmark the end of Pol II transcription units (Marchetti et al, 1998). Additionally, Pol II signals have been shown to be concentrated around active human U6 genes (Listerman et al, 2007), hinting at a regulatory relationship, thus supporting our hypothesis that the spatial arrangement of ADF-1L and 7SL-23 contributes to the high activity of 7SL-23. Lastly, it is noteworthy that all eight 7SL genes, ranging from 7SL-9 to 7SL-16, are all located in adjacent positions on chromosome 12 and show sensitivity to pathogen infection (Fig. 1B,C), highlighting a potential correlation between genomic location and the transcriptional activity of 7SL genes.

Based on existing literature regarding the domestication of PIF/harbinger TE-derived genes containing MADF domains, we propose that ADF-1L originated from PIF/harbinger TEs. Analysis of DNA transposon features revealed a sequence upstream of ADF-1L that aligns with PIF/harbinger characteristics, further supporting this derivation (Fig. 1I). The distribution of MADF family genes across various taxa, excluding mammals, reinforces the notion of their PIF/harbinger lineage (Shukla et al, 2014; Zimmermann et al, 2006). Given that Myb/SANT-domain proteins from PIF/harbinger TEs can regulate their DNA sequences, we hypothesize that ADF-1L may similarly regulate the transcription of nearby 7SL genes, categorizing ADF-1L as a "cis-acting factor" (Fig. 1J). Comparison of the MADF domain sequences and tertiary structures between *M. miiuy* ADF-1L (mmiADF-1L) and Drosophila ADF-1 (dmeADF-1) shows that, despite significant sequence differences, both have retained a helix-turn-helix (HTH) motif and a conserved three-dimensional structure (Fig. 1K; as such, mmiADF-1L likely maintains DNA-binding activity and functions as a transcription factor akin to dmeADF-1. Following pathogen infection, mmiADF-1L expression was significantly elevated, paralleling that of 7SL (Fig. 1L). To further explore its regulatory capacity, we constructed a plasmid for mmiADF-1L overexpression, which resulted in increased 7SL expression levels (Fig. 1M). Conversely, siRNAs

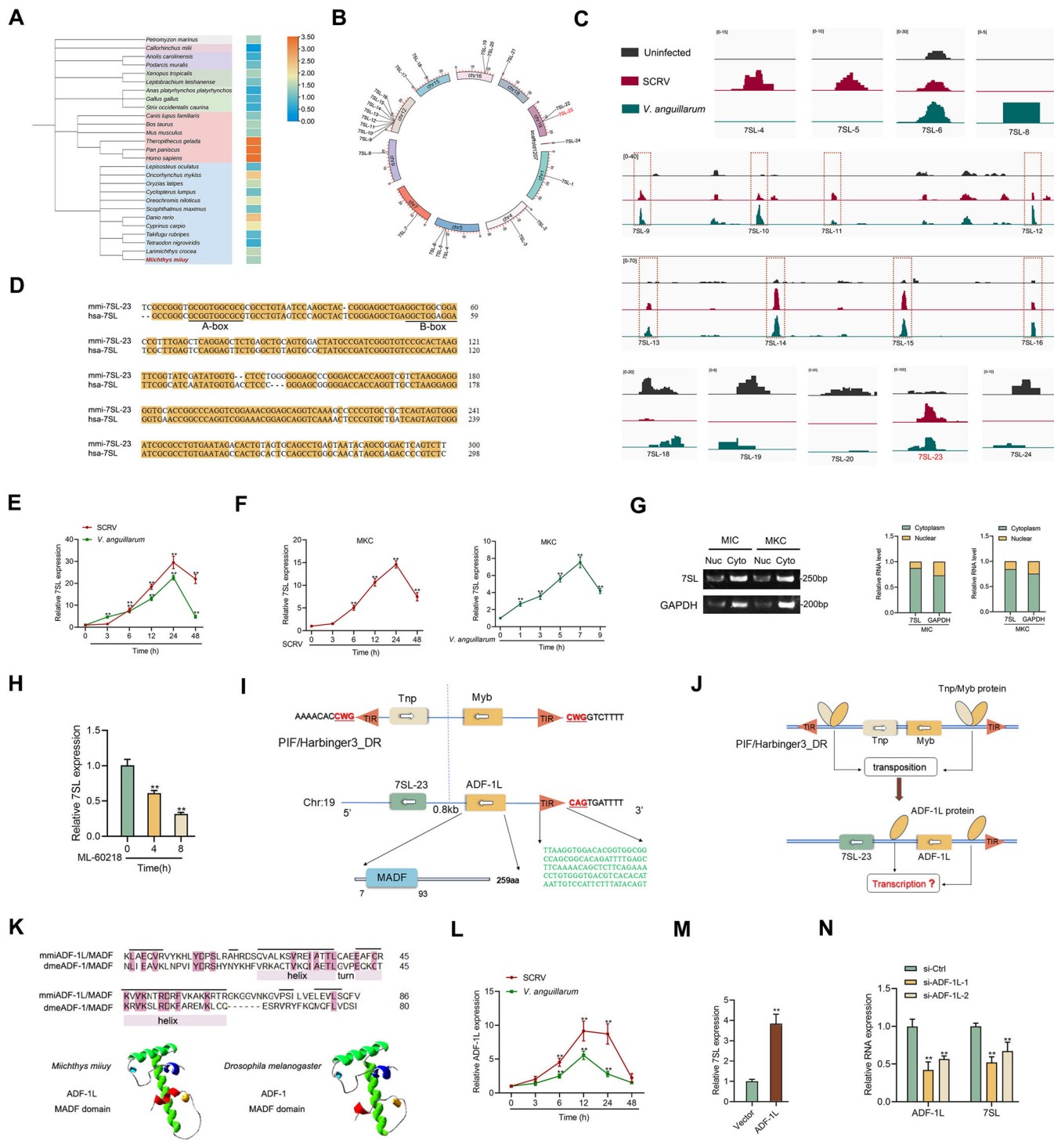

targeting ADF-1L significantly decreased the expression of both ADF-1L and 7SL (Fig. 1N). Previous studies showed that retrotransposons generally evolve under purifying selection in vertebrates, with proteins from most retrotransposon families showing a propensity to process their own RNA, exhibiting a cis-preference (Zhang et al, 2020). Therefore, proteins derived from transposons may have cis-preferences for regulating their own RNA/DNA sequences, contributing to the long-term preservation

of transposon sequences within host lineages, and this mechanism can also be used to explain the evolution of ADF-1L protein as a cis-acting factor to regulate 7SL-23.

## ADF-1L protein cis-regulates the adjacent 7SL-23 gene

Research on the MADF gene family is currently scarce, with most studies concentrating on Drosophila. In this lineage, MADF family

◄ 

**Figure 1.  Upstream of the active 7SL lies a PIF/harbinger TE-derived ADF-1L gene.**

(A) The number of 7SL gene copies in different vertebrates. Heatmap was constructed based on $\log_{10}$ (number). (B) The distribution of 7SL genes on chromosomes of *M. miiuy*. (C) The expression peaks of active 7SL genes in SCRV and *V. anguillarum* infected-spleen of *M. miiuy* detected by RNA-seq (GenBank: PRJNA819945 and PRJNA846999). IGV software was used for visualization, and the number in the upper left corner represents the sequencing readings of each 7SL gene. (D) Sequence alignment between *M. miiuy* 7SL-23 (mmi-7SL-23) and Homo sapiens 7SL (hsa-7SL). (E) Expression of 7SL in the spleen measured by qPCR at the indicated time after SCRV and *V. anguillarum* infection. **$P < 0.01$, values compared to 0 h group by one-way ANOVA with Dunnett's multiple comparisons test. SCRV: 6 h—$P = 0.0007$, 12 h—$P < 0.01$, 24 h—$P < 0.0001$, 48 h—$P < 0.0001$. *V. anguillarum*: 3 h—$P = 0.0002$, 6 h—$P < 0.0001$, 12 h—$P < 0.0001$, 24 h—$P < 0.0001$, 48 h—$P = 0.0001$. (F) Expression of 7SL in the MKC cells measured by qPCR at the indicated time after SCRV and *V. anguillarum* infection. **$P < 0.01$, values compared to 0 h group by one-way ANOVA with Dunnett's multiple comparisons test. SCRV: 6 h—$P < 0.0001$, 12 h—$P < 0.0001$, 24 h—$P < 0.0001$, 48 h—$P < 0.0001$. *V. anguillarum*: 1 h—$P = 0.0007$, 3 h—$P < 0.0001$, 5 h—$P < 0.0001$, 7 h—$P < 0.0001$, 9 h—$P < 0.0001$. (G) 7SL was mainly localized in the cytoplasm of MIC and MKC cells. (H) The transcription of 7SL was inhibited by a Pol III specific inhibitor (ML-60218). **$P < 0.01$, values compared to 0 h group by Student's t test. 4 h—$P = 0.0018$, 8 h—$P = 0.0002$. (I) The Harbinger3_DR transposon-derived ADF-1L is upstream of the active 7SL gene 7SL-23. (J) An evolution genetic model for ADF-1L regulating the 7SL-23 in a cis-acting manner. (K) Sequence alignment and three-dimensional structure of MADF domain of *M. miiuy* ADF-1L (mmiADF-1L) and *D. melanogaster* ADF-1L (dmeADF-1). The horizontal lines represent the predicted helix structures. (L) Expression of ADF-1L in spleen measured by qPCR at the indicated time after SCRV and *V. anguillarum* infection. **$P < 0.01$, values compared to 0 h group by one-way ANOVA with Dunnett's multiple comparisons test. SCRV: 6 h—$P = 0.0026$, 12 h—$P < 0.0001$, 24 h—$P < 0.0001$. *V. anguillarum*: 6 h—$P = 0.0010$, 12 h—$P < 0.0001$, 24 h—$P = 0.0002$. (M) qPCR analysis of 7SL levels in MKC cells transfected with ADF-1L for 48 h. **$P = 0.0007$, values compared to vector group by Student's t test. (N) qPCR analysis of 7SL levels in MKC cells transfected with si-ADF-1L. **$P < 0.01$, values compared to si-Ctrl group by Student's t test. ADF-IL: si-1, $P = 0.0023$. si-2, $P = 0.0021$. 7SL: si-1, $P = 0.0005$. si-2, $P = 0.0098$. All results are represented as the means ± SE of $n = 3$ biological replicates. Source data are available online for this figure.

genes have expanded, and all members feature a DNA-binding MADF domain at their N-terminus, indicating their potential role as transcription factors. Notably, ADF-1, a prominent gene in this family, serves as a transcriptional activator of the Drosophila alcohol dehydrogenase gene (Adh) by recognizing repeated trinucleotide motifs (England et al, 1990). Using this motif and predictions from online databases, we identified two potential binding motifs for mmiADF-1L (7SL-motif1 and 7SL-motif2) in the upstream and internal sequences of 7SL-23 (Fig. 2A). We identified all 19 7SL genes with over 90% similarity to the internal sequence of 7SL-23, retaining key A-box and B-box sequences essential for transcriptional activity (Müller and Benecke, 1999). Notably, the two motifs present in 7SL-23 were found to be unique to this gene (Fig. EV1A). Primers designed for chromatin immunoprecipitation (ChIP) confirmed that mmiADF-1L specifically binds to these motifs in the 7SL-23 gene (Fig. 2B). Additionally, we designed ChIP primers containing terminal inverted repeat (TIR) sequences located in the ADF-1L promoter region and found that ADF-1L can bind to its own promoter (Fig. 2B). Subsequently, we designed two 20 bp biotin-labeled probes containing the mmiADF-1L binding sites and obtained purified mmiADF-1L protein through recombinant methods (Fig. 2C,D). DNA pull-down assays revealed that the biotin-labeled probes effectively captured mmiADF-1L, while no products were obtained with non-biotin-labeled probes (Fig. 2E). Electrophoresis mobility shift assays (EMSA) using the two biotin-labeled probes displayed clear complexes, which were inhibited by excess cold DNA probes, confirming the direct interaction between mmiADF-1L and the two predicted binding sites in the upstream and internal sequences of 7SL-23 (Fig. 2F). However, we did not identify a specific motif in the ADF-1L promoter sequence that directly binds to the ADF-1L protein. Despite locating a sequence with two similar ADF-1 binding motifs (with one of the four conserved "G" residues in the motif deviating from being a "G") within the ADF-1L promoter, EMSA analysis revealed no binding to ADF-1L (Fig. EV1B,C). To further investigate ADF-1L's cis-regulatory specificity for 7SL-23, we examined its regulation of other 7SL genes. Due to the high sequence conservation between different copies of 7SL (Fig. EV1A), the designed RT primers may

have off target effects, making it impossible to directly measure the effect of ADF-1L knockdown on expression of various 7SL RNAs. Given that 7SL RNA transcription depends on RNA polymerase III and the TFIIIC complex, so we used ChIP assays to evaluate TFIIIC binding to the promoter sequences of various 7SL genes, including ATF binding sites essential for 7SL transcription (Müller and Benecke, 1999). Our results reveal that TFIIIC63 and TFIIIC90 bind to 7SL-23, as well as to 7SL-16 and 7SL-18, but not to 7SL-17, with the strongest interaction observed at 7SL-23 (Fig. 2G). Notably, ADF-1L knockdown specifically reduces the binding affinity between TFIIIC complex and the 7SL-23, without affecting 7SL-16 or 7SL-18, highlighting the selective regulation of 7SL-23 by ADF-1L (Fig. 2H). While 7SL-16 is significantly induced by pathogen infection like 7SL-23, it is not regulated by ADF-1L, suggesting that, although proximity to ADF-1L may influence regulation, other factors also contribute to the transcriptional response. These findings firmly establish ADF-1L as a cis-acting factor that regulates the 7SL-23 gene.

## Recruitment of KAT2B by ADF-1L promotes acetylation levels at ADF-1L and 7SL-23 gene loci

Previous studies have shown that MADF domains, along with its possible source Myb/SANT domain, are closely associated with chromatin remodeling and histone modification (Boyer et al, 2002; Yi et al, 2009), thus guiding the current investigation into the potential epigenetic molecular mechanism through which ADF-1L protein modulates 7SL-23 transcription. In Arabidopsis thaliana, the ALP2 protein, which acts as an antagonist of heterochromatin, is an evolved version of the PIF/harbinger transposase DNA-binding protein (Velanis et al, 2020). It contains the Myb/SANT domain and has taken on a new role as a component of PRC2, which regulates histone H3K27me3 methylation (Velanis et al, 2020). However, our study found no evidence that ADF-1L is involved in histone methylation, as it did not interact with PRC2 core components EZH2 and SUZ12 (Fig. 3A). Furthermore, the SANT domain is also considered a unique histone-interaction module that couples histone binding to acetylase catalysis in recent years (Boyer et al, 2004). For instance, kinetic analyses of a dimeric

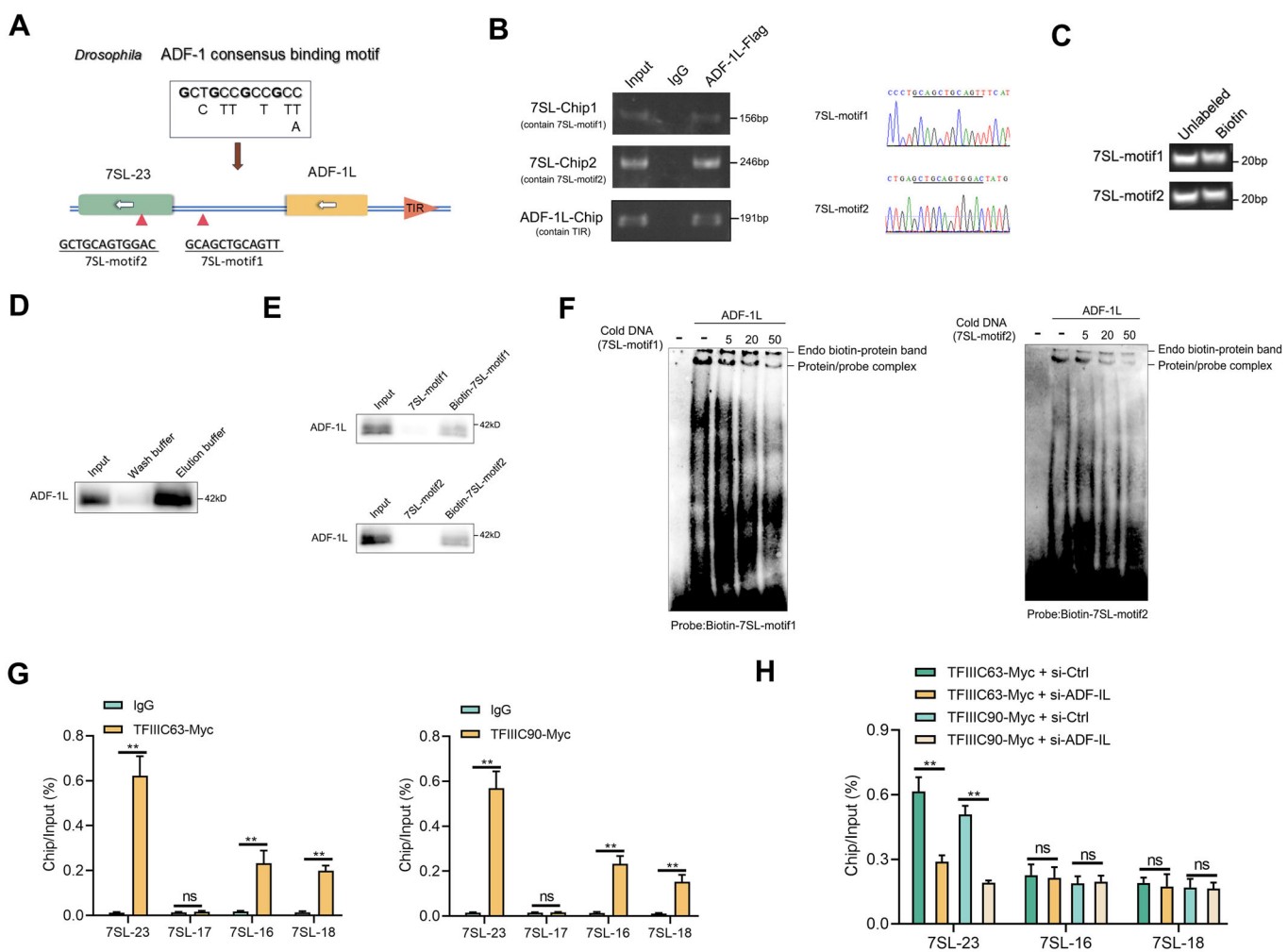

**Figure 2. ADF-1L protein cis-regulates the adjacent 7SL-23 gene.**

(A) Two ADF-1 binding motifs were predicted in the upstream and internal sequences of 7SL-23. (B) ChIP analysis of 7SL-23 and ADF-IL in MKC cells transfected with ADF-1L-Flag. (C) Unlabeled and biotin-labeled probes containing predicted ADF-1L binding sites obtained through PCR annealing. (D) ADF-1L protein obtained through recombinant purification. (E) DNA pull-down assays from lysates of MKC cells transfected with Flag-ADF-1L plasmids. (F) EMSA with purified ADF-1L, and biotin-labeled probes (7SL-motif1 and 7SL-motif2) with corresponding cold probes. (G) ChIP analysis of 7SL-23, 7SL-17, 7SL-16, and 7SL-18 in MKC cells transfected with TFIIIC63 or TFIIIC90 plasmids. **$P < 0.01$, ns, $P > 0.05$, values compared to IgG group by Student's $t$ test. TFIIIC63: 7SL-23, $P = 0.0003$. 7SL-17, $P = 0.5088$. 7SL-16, $P = 0.0028$. 7SL-18, $P = 0.0002$. TFIIIC90: 7SL-23, $P = 0.0002$. 7SL-17, $P = 0.3404$. 7SL-16, $P = 0.0004$. 7SL-18, $P = 0.0013$. (H) ChIP analysis of 7SL-23, 7SL-16, and 7SL-18 in MKC cells transfected with TFIIIC63 or TFIIIC90 plasmids and si-ADF-IL. **$P < 0.01$, ns, $P > 0.05$, values compared to si-Ctrl group by Student's $t$ test. TFIIIC63: 7SL-23, $P = 0.0014$. 7SL-16, $P = 0.7821$. 7SL-18, $P = 0.6669$. TFIIIC90: 7SL-23, $P = 0.0002$. 7SL-16, $P = 0.7583$. 7SL-18, $P = 0.8560$. All results are represented as the means ± SE of $n = 3$ biological replicates. Source data are available online for this figure.

GCN5-Ada2 complex showed that Ada2 significantly enhances the catalytic efficiency of GCN5 HAT on various histone substrates by factors of 10-fold, 31-fold, and 34-fold, respectively; these enhancements are abolished by a minor deletion within the SANT domain (Boyer et al, 2002). Similarly, KAT2A and KAT2B, histone acetyltransferases involved in eukaryotic transcription, displayed weak and strong binding to ADF-1L, respectively (Fig. 3B). ChIP analysis revealed that KAT2B specifically binds to the promoter of 7SL-23 and ADF-IL genes, while KAT2A does not (Fig. 3C). Furthermore, knocking down ADF-1L disrupted the binding of KAT2B to 7SL-23 and ADF-1L gene, impairing KAT2B's regulation of 7SL (Fig. 3D,E). Investigation into the impact of KAT2B on the regulation of ADF-1L and 7SL involved the construction of a

KAT2B-ΔHAT plasmid lacking acetyltransferase activity. The results indicated that the removal of the HAT domain significantly weakened KAT2B-ADF-1L interactions (Fig. 3F), and further revealed that KAT2B, rather than KAT2B-ΔHAT, induced the nuclear translocation of ADF-1L and the high expression of 7SL and ADF-1L (Fig. 3G,H). Additionally, ADF-1L-ΔMADF plasmids lacking the MADF domain were generated, and the results demonstrated that ADF-1L, rather than ADF-1L-ΔMADF, promoted the transcription of 7SL (Fig. 3I,J). Further evidence suggested that ADF-1L binds to KAT2B and promotes its transcription of 7SL in a MADF domain-dependent manner (Fig. 3K,L). Finally, acetylation modifications were indeed present at the 7SL-23 and ADF-1L gene loci, and it was found that ADF-1L,

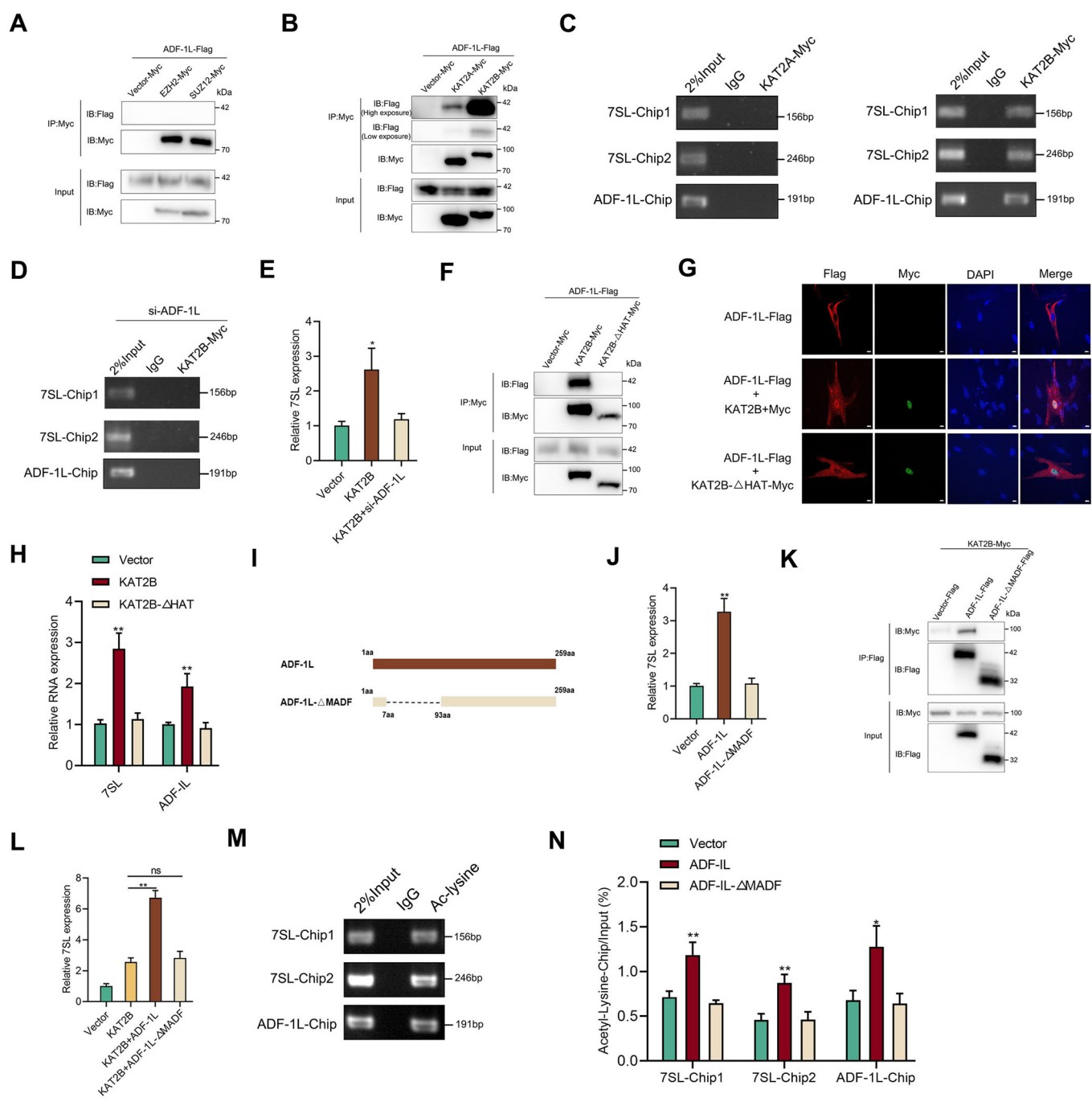

**Figure 3. Recruitment of KAT2B by ADF-1L promotes acetylation levels at ADF-1L and 7SL-23 gene loci.**

(A) Immunoprecipitation analysis of ADF-1L in EPC cells transfected with EZH2-Myc or SUZ12-Myc. (B) Immunoprecipitation analysis of ADF-1L in EPC cells transfected with KAT2A-Myc or KAT2B-Myc. (C) ChIP analysis of 7SL-23 and ADF-IL in MKC cells transfected with KAT2A or KAT2B. (D) ChIP analysis of 7SL-23 and ADF-IL in MKC cells co-transfected with KAT2B-Myc plasmids and si-ADF-1L. (E) qPCR analysis of 7SL in MKC cells co-transfected with KAT2B plasmids and si-ADF-1L. *P = 0.0110, values compared to vector group by Student's t test. (F) Immunoprecipitation analysis of ADF-1L in EPC cells transfected with KAT2B-Myc or KAT2B-ΔHAT-Myc. (G) MKC cells were transfected with indicated plasmids, then samples were subjected to a fluorescence microscope analysis. Scale bars, 20 μm. (H) qPCR analysis of 7SL in MKC cells transfected with KAT2A or KAT2B. **P < 0.01, values compared to vector group by Student's t test. 7SL, P = 0.0013. ADF-IL, P = 0.072. (I) Schematic description of the ADF-1L truncated mutant (ADF-1L-ΔMADF). (J) qPCR analysis of 7SL in MKC cells transfected with ADF-1L or ADF-1L-ΔMADF. **P = 0.0007, values compared to vector group by Student's t test. (K) Immunoprecipitation analysis of KAT2B in EPC cells transfected with ADF-1L or ADF-1L-ΔMADF. (L) ADF-1L instead of ADF-1L-ΔMADF enhanced the transcription of 7SL by KAT2B. **P = 0.0002, ns, P = 0.4262, values were determined by Student's t test. (M) ChIP analysis of 7SL-23 and ADF-IL by Ac-lysine antibody in MKC cells. (N) ChIP analysis of 7SL-23 and ADF-IL by Ac-lysine antibody in MKC cells transfected with ADF-IL or ADF-1L-ΔMADF. **P < 0.01, *P < 0.05, values compared to vector group by Student's t test. 7SL-Chip1, P = 0.0069. 7SL-Chip2, P = 0.0036. ADF-IL-Chip, P = 0.0163. All results are represented as the means ± SE of n = 3 biological replicates. Source data are available online for this figure.

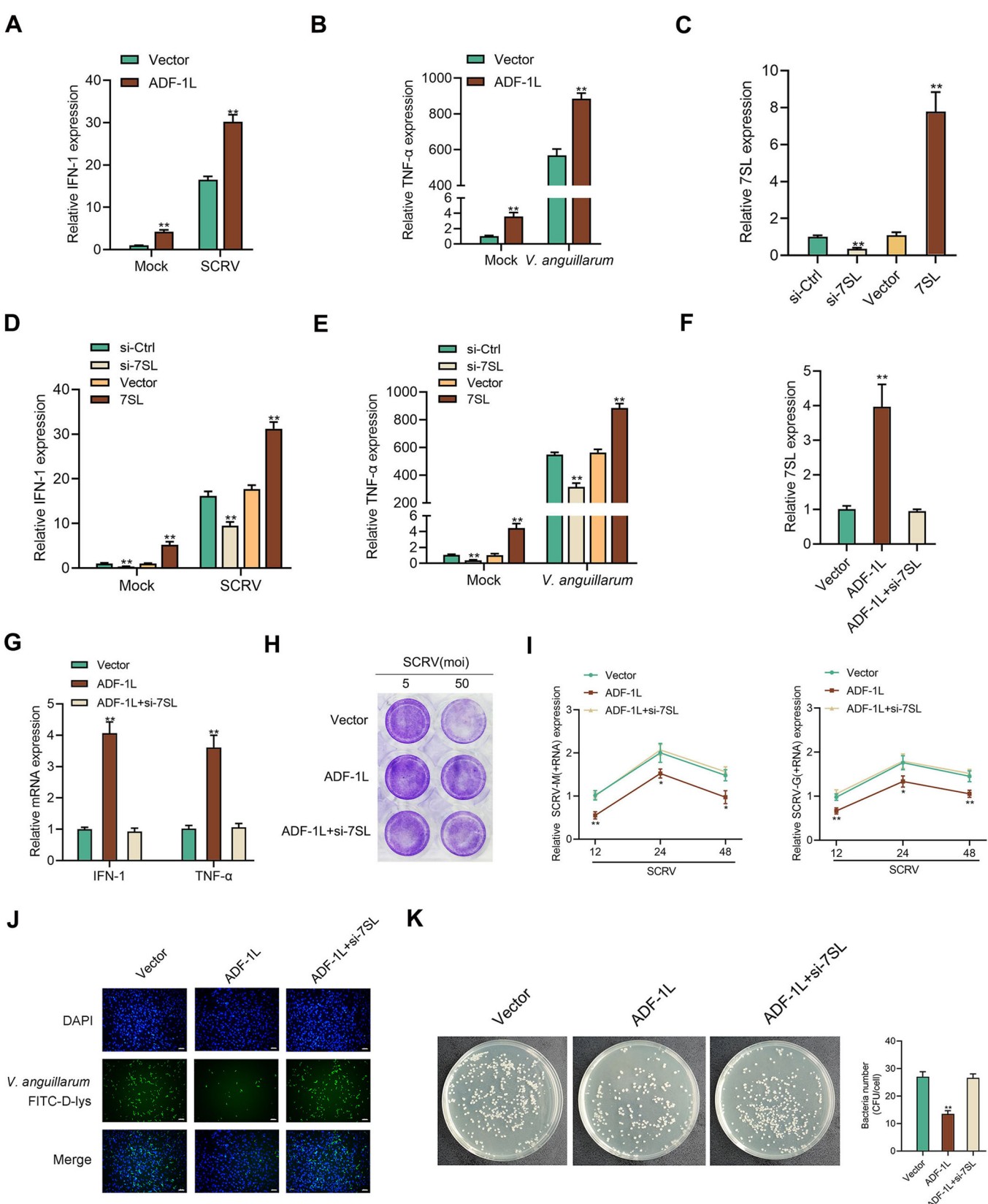

◀ **Figure 4.  ADF-1L promotes immunity by regulating the 7SL.**

(A) qPCR analysis of IFN-1 expression in MKC cells transfected ADF-1L plasmids and then infected with SCRV for 24 h. **$P < 0.01$, values compared to vector group by Student's *t* test. Mock, $P = 0.0002$. SCRV, $P = 0.0002$. (B) qPCR analysis of TNF-α expression in MKC cells transfected ADF-1L plasmids and then infected with *V. anguillarum* for 6 h. **$P < 0.01$, values compared to vector group by Student's *t* test. Mock, $P = 0.0012$. *V. anguillarum*, $P = 0.0003$. (C) qPCR analysis of 7SL in MKC cells transfected with indicated si-7SL or 7SL plasmids. **$P < 0.01$, values compared to control group by Student's *t* test. si-7SL, $P = 0.0012$. 7SL, $P = 0.0004$. (D) qPCR analysis of IFN-1 in MKC cells transfected with si-7SL or 7SL plasmids and then infected with SCRV for 24 h. **$P < 0.01$, values compared to control group by Student's *t* test. Mock: si-7SL, $P = 0.0023$. 7SL, $P = 0.0005$. SCRV: si-7SL, $P = 0.0009$. 7SL, $P = 0.0002$. (E) qPCR analysis of TNF-α in MKC cells transfected with si-7SL or 7SL plasmids and then infected with *V. anguillarum* for 6 h. **$P < 0.01$, values compared to control group by Student's *t* test. Mock: si-7SL, $P = 0.0007$. 7SL, $P = 0.0006$. *V. anguillarum*: si-7SL, $P = 0.0002$. 7SL, $P = 0.0001$. (F, G) Relative expression of 7SL, IFN-1, and TNF-α in MKC cells transfected with ADF-1L or ADF-1L + si-7SL. **$P < 0.01$, values compared to control group by Student's *t* test. 7SL, $P = 0.0014$. IFN-1, $P = 0.0001$. TNF-α, $P = 0.0004$. (H) Virus spot analysis in MKC cells transfected with ADF-1L or ADF-1L + si-7SL. (I) qPCR analysis of SCRV-M and SCRV-G RNA levels in MKC cells transfected with ADF-1L or ADF-1L + si-7SL and then infected with SCRV for 12, 24, or 48 h. **$P < 0.01$, *$P < 0.05$, values compared to vector group by Student's *t* test. SCRV-M: 12 h, $P = 0.0042$. 24 h, $P = 0.0268$. 48 h, $P = 0.0110$. SCRV-G: 12 h, $P = 0.0056$. 24 h, $P = 0.0217$. 48 h, $P = 0.0094$. (J) MKCs were transfected with ADF-1L or ADF-1L + si-7SL, then infected with FITC-labeled *V. anguillarum*, and then examined by using a fluorescence microscope. Scale bars, 20 μm. (K) Flat plate coating analysis of intracellular bacterial number in MKCs transfected with ADF-1L or ADF-1L + si-7SL and then infected with *V. anguillarum*. **$P = 0.0004$, values compared to vector group by Student's *t* test. All results are represented as the means ± SE of $n = 3$ biological replicates. Source data are available online for this figure.

rather than ADF-1L-ΔMADF, can increase the acetylation levels at these loci (Fig. 3M,N). In summary, ADF-1L recruits KAT2B to facilitate its own nuclear translocation and enhance the acetylation levels of ADF-IL and 7SL-23 genes as a cis-acting factor. These results support the view that the MADF domain plays a key role in the regulation of histone acetylation. Given the high conservation of KAT from yeast to mammals, our study also offers valuable insights into the functions of the MADF gene family.

## ADF-1L promotes immunity by regulating the 7SL

Given the increased expression of ADF-1L and 7SL after pathogen infection, we explored their potential roles in immune response. ADF-1L overexpression was shown to boost the levels of IFN-1 and TNF-α during SCRV or *V. anguillarum* infection, suggesting it positively regulates the immune system (Fig. 4A,B). The si-7SL was able to inhibit 7SL expression, while the 7SL overexpression plasmid notably enhanced 7SL expression levels (Fig. 4C). Similar to ADF-1L, knockdown or overexpression of 7SL can significantly inhibit or promote the expression levels of TNF-α and IFN-1 (Fig. 4D,E), indicating that both 7SL RNA and ADF-1L protein positively regulate immunity. Considering ADF-1L as an upstream transcription factor for 7SL, we investigated whether ADF-1L's immune function relies on its regulation of 7SL. Our findings revealed that si-7SL restored the elevated IFN-1 and TNF-α levels in ADF-1L-overexpressed MKC cells (Fig. 4F,G). Furthermore, ADF-1L overexpression reduced viral plaque formation and inhibited SCRV replication, with si-7SL reversing the enhanced resistance to SCRV infection (Fig. 4H,I). Using FITC-D-Lys to label *V. anguillarum* and conducting plate counting, we found that ADF-1L overexpression impeded *V. anguillarum* invasion, and knocking down 7SL under ADF-1L-overexpressed conditions similarly reduced antibacterial defense (Fig. 4J,K). Overall, these results indicate that ADF-1L promotes immunity and inhibits viral replication and bacterial infection via 7SL regulation. Notably, previous studies have linked the MADF gene family to housekeeping genes, such as Drosophila CP190, an essential transcription factor for pol III-transcribed gene promoters and insulators, and CP60, which binds chromatin as a key CP190 partner (Melnikova et al, 2023). Our study provides further evidence of the relationship between the MADF family and pol III genes, revealing ADF-1L's dual roles in immunity and gene regulation.

## ADF-1L regulates the 7SL/miR-2187-3p/TRAF6 signaling axis

Originally deemed a housekeeping gene, the 300-nt long ncRNA 7SL forms a ribonucleoprotein complex (RNP) with six signal-recognition proteins (SRPs), thus binding signal sequences of secretory and transmembrane proteins to facilitate their targeting to the membrane translocation apparatus in the endoplasmic reticulum (Weigert et al, 2023). Recently, the role of 7SL in cancer and immune processes has garnered attention (Abdelmohsen et al, 2014; Ouyang et al, 2016; Zhang et al, 2023). 7SL can bind the 3′ UTR of TP53 mRNA, encoding the tumor suppressor, and then leading to the suppression of p53 translation (Abdelmohsen et al, 2014). Moreover, 5′triphosphorylated 7SL RNA serves as a direct ligand for the RIG-I to mediate the IFN response (Zhang et al, 2023). In this study, we have elucidated an additional molecular pathway through which 7SL functions as a competing endogenous RNA (ceRNA) in immunity. RNA pull-down assays results showed that Ago2, one protein that can form the Ago2/RNA-induced silencing complex (RISC) to regulate the miRNA (Chendrimada et al, 2005), could be enriched by biotin-labeled 7SL but not unlabeled 7SL (Fig. 5A). NARL, a previously identified lncRNA that binds Ago2 (Zheng et al, 2021), served as a positive control (Fig. 5A). RIP assays further confirmed that 7SL is significantly enriched in Ago2-containing complexes, indicating its potential to bind miRNAs (Fig. 5B). Next, we used TargetScan and miRanda software to identify miRNAs targeting 7SL, discovering three candidates: miR-133-3p, miR-2187-3p, and miR-29a-3p (Fig. 5C). Of these, miR-2187-3p exhibited notably lower expression compared to the others during 7SL overexpression, and its expression significantly increased upon 7SL silencing (Fig. 5D). To assess whether 7SL influences miR-2187-3p activity, we developed a miR-2187-3p sensor and demonstrated that 7SL sponges miR-2187-3p, reducing luciferase activity (Fig. 5E). The luciferase assay confirmed that miR-2187-3p can inhibit the activity of the wild-type 7SL plasmid but not the mutant form lacking the binding sites (Fig. 5F). To further establish the direct interaction between 7SL and miR-2187-3p, we constructed plasmids recognizing lncRNA using the MS2 protein, which allowed us to pull down miR-2187-3p with the GFP-MS2-12X complex. qPCR analysis indicated that miR-2187-3p was significantly more enriched in wild-type 7SL than in the mutant (Fig. 5G). A pull-down assay with

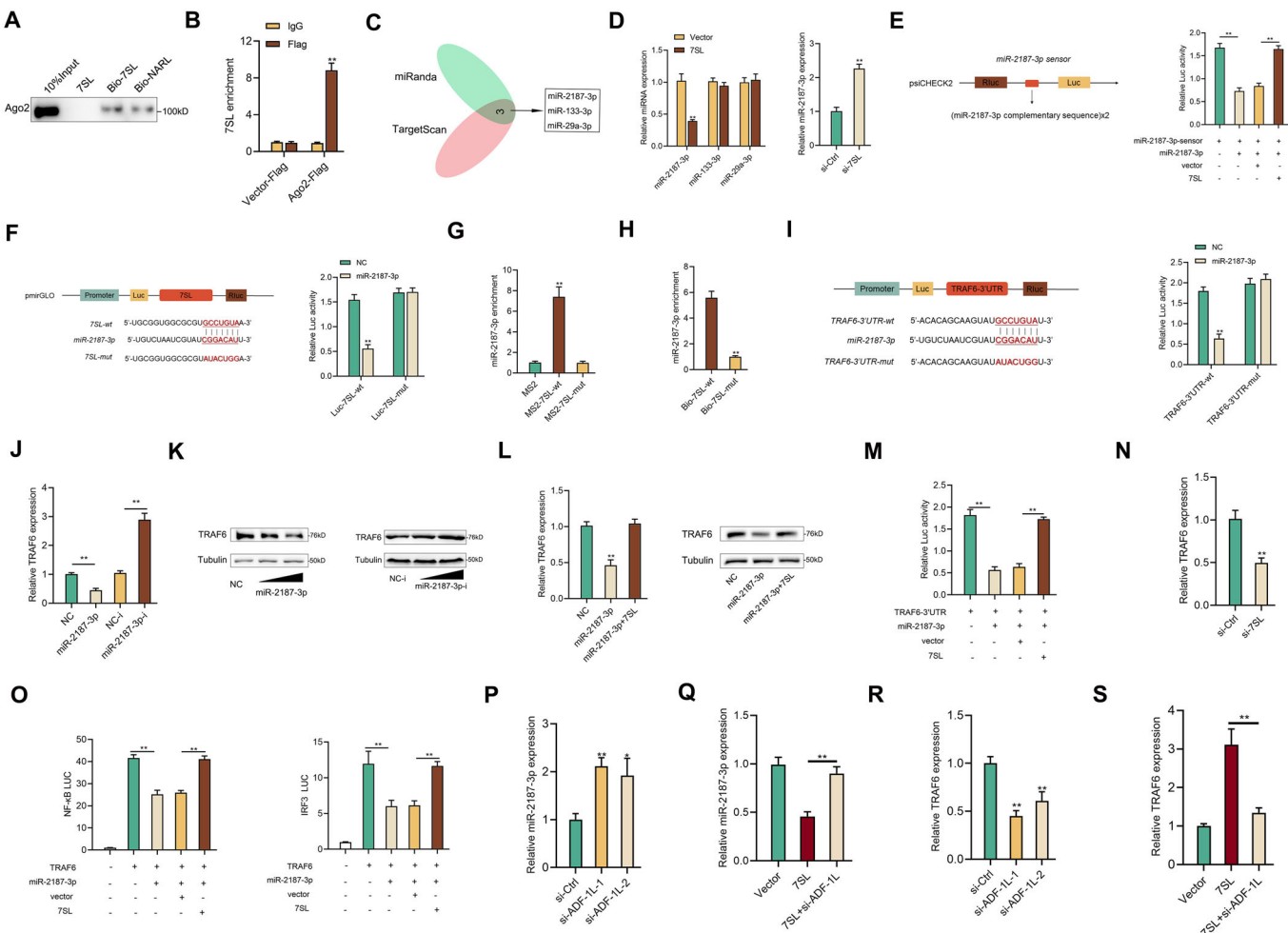

**Figure 5. ADF-1L regulates the 7SL/miR-2187-3p/TRAF6 signaling axis.**

(A) RNA pull-down analysis for cell lysates (input), unlabeled 7SL probe (negative control) pull down, biotin-labeled 7SL probe pull down (experimental group) and NARL probe pull down (positive control). (B) The Ago2-RIP assay for the amount of 7SL in MKCs transfected Ago2-Flag or pcDNA3.1-Flag. **$P < 0.0001$, values compared to IgG group by Student's $t$ test. (C) A schematic illustration showing overlapping of the target miRNAs of 7SL predicted by TargetScan and miRanda. (D) Relative expression of candidate miRNAs in MKCs transfected with 7SL plasmids or si-7SL. **$P < 0.01$, values compared to control group by Student's $t$ test. 7SL: $P = 0.0006$. si-7SL: $P = 0.0002$. (E) The luciferase activity of constructed miR-2187-3p sensor in MKC cells transfected with indicated mimics and plasmids. **$P < 0.01$, values compared to control group by Student's $t$ test. miR-2187-3p: $P = 0.0001$. 7SL: $P = 0.0001$. (F) The luciferase activity of Luc-7SL in MKC cells transfected with miR-2187-3p. miR-2187-3p-binding sites in 7SL wild-type form (Luc-7SL-wt) and the mutated form (Luc-7SL-mut) were shown. **$P = 0.0002$, values compared to control group by Student's $t$ test. (G) MS2-RIP method was used to identify the binding between 7SL-wt or 7SL-mut and miR-2187-3p in MKC cells. **$P = 0.0003$, values compared to MS2 group by Student's $t$ test. (H) RNA pull-down assay to detect the binding between 7SL-wt or 7SL-mut and miR-2187-3p. **$P < 0.0001$, values compared to control group by Student's $t$ test. (I) The luciferase activity of TRAF6 3′UTR in MKC cells transfected with miR-2187-3p. miR-2187-3p binding sites in wild-type of TRAF6 3′UTR (TRAF6-3′UTR-wt) and a mutated form of 3′UTR (TRAF6-3′UTR-mut) were shown. **$P = 0.0002$, values compared to control group by Student's $t$ test. (J, K) The expression of TRAF6 in MKC cells transfected with indicated mimics was determined by qPCR (J) and Western blotting (K). **$P < 0.01$, values compared to control group by Student's $t$ test. miR-2187-3p, $P = 0.0003$. miR-2187-3p, $P = 0.0002$. (L) mRNA and protein levels of TRAF6 in MKC cells transfected with miR-2187-3p or miR-2187-3p + 7SL plasmid. **$P = 0.0005$, values compared to NC group by Student's $t$ test. (M) Luciferase activity of TRAF6-3′UTR in MKC cells transfected with indicated mimics and plasmids. **$P < 0.01$, values compared to control group by Student's $t$ test. miR-2187-3p, $P = 0.0001$. 7SL, $P < 0.0001$. (N) qPCR analysis of TRAF6 in MKC cells transfected with si-7SL. **$P = 0.0003$, values compared to si-Ctrl group by Student's $t$ test. (O) Luciferase activity of NF-κB and IRF3 report genes in MKC cells co-transfected with indicated mimics and plasmids. **$P < 0.01$, values compared to control group by Student's $t$ test. NF-κB: miR-2187-3p, $P = 0.0003$. 7SL, $P = 0.0003$. IRF3: miR-2187-3p, $P = 0.0061$. 7SL, $P = 0.0004$. (P, Q) qPCR analysis of miR-2187-3p in MKC cells transfected with si-ADF-IL (P) or with 7SL and si-ADF-IL(Q). **$P < 0.01$, *$P < 0.05$, values compared to control group by Student's $t$ test. si-1, $P = 0.0009$. si-2, $P = 0.0137$. 7SL, $P = 0.0009$. (R, S) qPCR analysis of TRAF6 in MKC cells transfected with si-ADF-IL (R) or with 7SL and si-ADF-IL(S). **$P < 0.01$, *$P < 0.05$, values compared to control group by Student's $t$ test. si-1, $P = 0.0004$. si-2, $P = 0.0042$. 7SL, $P = 0.0020$. All results are represented as the means ± SE of $n = 3$ biological replicates. Source data are available online for this figure.

biotinylated 7SL confirmed that miR-2187-3p could be pulled down by wild-type 7SL but not the mutant (Fig. 5H). miRNAs typically act as negative regulators of gene expression, binding to the 3′ UTR of target mRNAs. We found a complementary sequence for miR-

2187-3p in the 3′ UTR of TRAF6 and constructed both wild-type and mutant TRAF6 3′ UTR luciferase reporter plasmids. The luciferase assay showed that miR-2187-3p mimics significantly inhibited luciferase activity of the wild-type but not the mutant

(Fig. 5I). Additionally, miR-2187-3p mimics reduced TRAF6 expression, while inhibitors increased it (Fig. 5J,K), and 7SL counteracts miR-2187-3p's effects on TRAF6 mRNA and protein levels and si-7SL inhibits the mRNA expression of TRAF6 (Fig. 5L–N). Since TRAF6 activates NF-κB and IRF3, we evaluated the roles of 7SL and miR-2187-3p in regulating these reporter genes. Luciferase assays demonstrated that 7SL could counteract miR-2187-3p's negative impact on NF-κB and IRF3 activities (Fig. 5O). Finally, to test the biological relevance between ADF-IL and 7SL/miR-2187-3p/TRAF6 axis, we examined the expression of miR-2187-3p and TRAF6 after ADF-1L knockdown. We found that si-ADF-1L could enhance miR-2187-3p expression and rescue the inhibitory effect of 7SL on miR-2187-3p (Fig. 5P,Q). Consistently, si-ADF-1L reduced TRAF6 expression and inhibited the positive regulation of TRAF6 by 7SL (Fig. 5R,S). Together, these findings suggest that 7SL serves as a sponge for miR-2187-3p to enhance TRAF6 expression, thus regulating IRF3 and NF-κB signaling, and ADF-1L promotes this immune process by facilitating the transcription of 7SL.

# Methods

### Reagents and tools table

| Reagent/resource | Reference or source | Identifier or catalog number |
| --- | --- | --- |
| **Experimental models** | | |
| HEK293T cells | ATCC | RRID:CVCL_0063 |
| EPC cells | ATCC | RRID:CVCL_4361 |
| MKC cells | Laboratory of Fish Molecular Immunology (Shanghai, China) | N/A |
| MIC cells | Laboratory of Fish Molecular Immunology (Shanghai, China) | N/A |
| *Miichthys miiuy* | Laboratory of Fish Molecular Immunology (Shanghai, China) | N/A |
| **Recombinant DNA** | | |
| pcDNA3.1-ADF-IL | This study | N/A |
| pcDNA3.1-ADF-IL-6xHis | This study | N/A |
| pcDNA3.1-ADF-ILΔMADF | This study | N/A |
| pcDNA3.1-TFIIIC63 | This study | N/A |
| pcDNA3.1-TFIIIC90 | This study | N/A |
| pcDNA3.1-EZH2 | This study | N/A |
| pcDNA3.1-SUZ12 | This study | N/A |
| pcDNA3.1-KAT2A | This study | N/A |
| pcDNA3.1-KAT2B | This study | N/A |
| pcDNA3.1-KAT2BΔHAT | This study | N/A |
| pcDNA3.1-TRAF6 | This study | N/A |
| pcDNA3.1-Ago2 | This study | N/A |
| pcDNA3.1-7SL | This study | N/A |
| PmirGLO-7SL | This study | N/A |
| PmirGLO-7SL-mut | This study | N/A |

| Reagent/resource | Reference or source | Identifier or catalog number |
| --- | --- | --- |
| PmirGLO-TRAF6-3′UTR | This study | N/A |
| PmirGLO-TRAF6-3′UTR-mut | This study | N/A |
| MS2-7SL | This study | N/A |
| MS2-7SL-mut | This study | N/A |
| **Antibodies** | | |
| Anti-Flag | Beyotime | Cat# AF519 |
| Anti-Myc | Beyotime | Cat# AF2864 |
| Anti-GFP | Beyotime | Cat# AG281 |
| Anti-Tubulin | Beyotime | Cat# AT819 |
| Anti-Ago2 | Boster | Cat# BM4920 |
| Anti-TRAF6 | Boster | Cat# A00185 |
| Anti-Mouse IgG, HRP | Abbkine | Cat# A25012 |
| **Oligonucleotides and other sequence-based reagents** | | |
| Primers | This study | Table EV1 |
| siRNAs | GenePharma | N/A |
| **Chemicals, enzymes and other reagents** | | |
| Lipofectamine RNAiMAX | Invitrogen | Cat# 13778150 |
| Lipofectamine 3000 | Invitrogen | Cat# L3000015 |
| Ribo RNAmax-T7 Biotin Labeling Transcription Kit | RiboBio | Cat# C11002-2 |
| Ribo RNAmax-T7 kit | RiboBio | Cat# C11001-2 |
| PureBinding®RNA-Protein pull-down Kit | Geneseed | Cat# P0202 |
| EMSA kit | Beyotime | Cat# GS606 |
| Magna RIP RNA-Binding Protein Immunoprecipitation Kit | Millipore | Cat# 17-700 |
| BCA Protein Assay kit | Beyotime | Cat# P0012S |
| Endotoxin-Free Plasmid DNA Miniprep Kit | Tiangen | Cat# DP118 |
| Dual-Luciferase® Reporter Assay System | Promega | Cat# E1980 |
| FastQuant RT Kit | Tiangen | Cat# KR106-03 |
| SYBR Premix Ex Taq™ | Takara | Cat# DRR041S |
| TRIzol | Invitrogen | Cat#15596026CN |
| Mut Express II Fast Mutagenesis Kit V2 | Vazyme | Cat# C214 |
| DAPI | Beyotime | Cat# C1002 |
| Protein A/G agarose beads | Beyotime | Cat# P2012 |
| anti-His beads | Solarbio | Cat# M2300 |
| streptavidin agarose beads | Invitrogen | Cat# SA10004 |
| FITC-D-Lys | Xiamen Shengguang Biotechnology | Cat# I0201 |
| DNase I | Beyotime | Cat# D7076 |
| PVDF membranes | Millipore | Cat# IPVH00010 |

| Reagent/resource | Reference or source | Identifier or catalog number |
|---|---|---|
| **Software** | | |
| GraphPad Prism 8 | GraphPad Software | RRID:SCR_002798 |
| Integrative Genomics Viewer | SciCrunch Registry | RRID:SCR_011793 |
| **Other** | | |

## Ethics statement

All animal experimental procedures were performed following the National Institutes of Health's Guide for the Care and Use of Laboratory Animals, and the experimental protocols were approved by the Research Ethics Committee of Shanghai Ocean University (No. SHOU-DW-2018-047).

## Database mining and sequence analysis

Species tree was constructed by submitting species names to the NCBI (https://www.ncbi.nlm.nih.gov/Taxonomy/CommonTree/). Based on the high conservatism of the 7SL sequence, the 7SL sequence of *M. miiuy* was as a query to seek the genome database of different vertebrate species to determine their number of copies for 7SL. The protein domain structure was predicted by the SMART website (http://smart.embl-heidelberg.de/), followed by multiple comparisons using DNAman. The three-dimensional structures of the MADF domains were predicted using SWISS-MODEL Repository software (https://swissmodel.expasy.org/repository/). A collective database of transposable elements of fish (https://www.fishtedb.org/project/) was used to predict the transposon characteristic sequences in the DNA sequence of *M. miiuy*. The website (http://gene-regulation.com/) is used to predict the binding motifs of ADF-1 in the promoter of ADF-IL and 7SL-23 gene.

## Sample and challenge

*M. miiuy* weighing approximately 50 g was sourced from Zhoushan Fisheries Research Institute, Zhejiang Province, China. The fish underwent a 6-week acclimation period in aerated seawater tanks at 25 °C prior to the commencement of experiments. The experimental protocol for SCRV and *V. anguillarum* infection was conducted according to previous descriptions (Geng et al, 2022b).

## Cell culture and treatment

*M. miiuy* kindey cells (MKCs) and *M. miiuy* intestine cells (MICs) were cultured in L-15 medium (HyClone) supplemented with 15% fetal bovine serum (FBS; Gibco), 100 U/ml penicillin, and 100 μg/ml streptomycin at 26 °C. HEK293T cells were cultured in DMEM (HyClone) supplemented with 10% FBS, 100 U/ml penicillin, and 100 mg/ml streptomycin at 37 °C in 5% $CO_2$. All cells are routinely tested for mycoplasma. For stimulation experiments, Cells were challenged with SCRV at a multiplicity of infection (MOI) of 5 and then collected at various time points for RNA extraction.

## Plasmids construction

To construct the TRAF6 3′UTR reporter vector, the 3′UTR region of *M. miiuy* TRAF6 gene was amplified using PCR and cloned into the pmirGLO luciferase reporter vector (Promega). To construct 7SL luciferase genes, the sequence of 7SL in *M. miiuy* was cloned into the pmirGLO luciferase reporter vector. The mutated forms with point mutations in the miR-2187-3p binding site were synthesized using Mut Express II Fast Mutagenesis Kit V2 with specific primers. To construct the TRAF6 expression plasmid, the full length of the coding sequence region and 3′UTR of the TRAF6 gene were amplified by specific primer pairs and cloned into the pcDNA3.1 vector (Invitrogen). Also, 7SL expression plasmids were constructed by cloning the 7SL sequence region of *M. miiuy*. To construct the ADF-1L, TFIIIC63, TFIIIC90, EZH2, SUZ12, KAT2A, and KAT2B expression plasmid, the coding sequence region of the ADF-1L gene was amplified by specific primer pairs and cloned into pcDNA3.1 vector. Then, ADF-1LΔMADF and ADF-1L-6×His plasmids were generated by PCR on the basis of the ADF-1L plasmid. The correct construction of the recombinant plasmids was verified through Sanger sequencing and extracted using an Endotoxin-Free Plasmid DNA Miniprep Kit (Tiangen, China) before using plasmids. The primers are listed in Table EV1.

## Protein purification and analysis

(His6)-Flag-tagged ADF-1L of *M. miiuy* was transiently overexpressed in 293 T cells and lysed in a binding buffer (500 mM NaCl, 20 mM Tris/HCl [pH 7.4], 20 mM Imidazole, and 1% Triton X-100) supplemented with a protease inhibitor cocktail (Roche). The lysate was incubated overnight at 4 °C with anti-His beads (Solarbio). Subsequently, the anti-His beads were washed with a washing buffer (500 mM NaCl, 20 mM Tris/HCl [pH 7.4], 20 mM Imidazole, and 1% Triton X-100). ADF-1L proteins were eluted from the beads by the addition of elution buffer (500 mM NaCl, 20 mM Tris/HCl [pH 7.4], 500 mM Imidazole, and 1% Triton X-100). The purity of recombinant ADF-1L was assessed by SDS-PAGE separation.

## RNA interference

The 7SL-23-specific small interfering RNA (si-7SL) sequence is 5′-GGCGGACCGUUUGAGCUCATT-3′. The two ADF-1L-specific small interfering RNAs (si-ADF-1L-1 and si-ADF-1L-2) sequences are 5′-GGUGCAGAACUCAACAAUATT-3′ and 5′-GGUUUGUUGAAGUAAUCAATT-3′ respectively. The scrambled control RNA sequences were 5′-UUCUCCGAACGUGUCAC-GUTT-3′. The miR-2187-3p mimics are synthetic double-stranded RNAs (dsRNAs) with stimulating naturally occurring mature miRNAs. The miR-2187-3p mimics sequence was 5′-UUACAGGCUAUGCUAAUCUGU-3′. The negative control mimics sequence was 5′-UUCUCCGAACGUGUCACGUTT-3′. miRNA inhibitors are synthetic single-stranded RNAs (ssRNAs) that sequester intracellular miRNAs and block their activity in the RNA-interfering pathway. The miR-2187-3p inhibitors sequence was 5′-ACAGAUUAGCAUAGCCUGUAA-3′. The negative control inhibitor sequence was 5′-CAGUACUUUUGUGUAGUA-CAA-3′.

## Cells transfection

Cells were transiently transfected with siRNA in 24-well plates using Lipofectamine RNAiMAX (Invitrogen), while transfection of DNA plasmids was conducted using Lipofectamine 3000 (Invitrogen) following the manufacturer's protocols. For functional assessments, the overexpression plasmid (500 ng per well) and siRNA (100 nM) were transfected into cells in a culture medium and subsequently harvested for further analysis.

## Virus yield quantification

For stimulation experiments, Cells were challenged with SCRV at a multiplicity of infection (MOI) of 5 and harvested at different times for RNA extraction. The replication of SCRV was detected by qPCR. For virus plaque assay, the cell monolayer was washed with PBS, fixed with 4% paraformaldehyde, and stained with 1% crystal violet.

## Bacteria invasion

*V. anguillarum* was subjected to three washes with fresh medium, followed by centrifugation and dilution in fresh medium containing 100 mM FITC-D-Lys (Shengguang Biotechnology) for 30 min. at 37 °C. FITC-labeled *V. anguillarum* was introduced to the dishes in 200 μl of serum-free L-15 medium, allowing for infection of MKC cells at different time points. The infected cells underwent three washes with PBS, followed by the addition of 200 μl of L-15 medium supplemented with 15% FBS (Life Technologies), 100 U/ml penicillin, and 100 mg/ml streptomycin at 26 °C for 1 h. Subsequently, the infected cells were washed three times with PBS and fixed by incubating with 0.2% Triton X-100 in 4% paraformaldehyde for 30 min. at room temperature. Finally, the cells were imaged under a Leica DMiL8 fluorescence microscope.

## Colony formation assay

Experimental procedures for *V. anguillarum* infection were performed as described (Geng et al, 2022b). Briefly, *V. anguillarum* infected MKC cells were washed three times with PBS, and 200 ml of L-15 medium supplemented with 15% FBS (Life Technologies),100 U/ml penicillin, and 100 mg/ml streptomycin were added at 26 °C for 1 h. Then 1 ml sterile ddH$_2$O was added to lyse cells 4 °C for 30 min. Then the above cell lysate was coated, incubated on the agar plates at 37 °C for 12 h, and the colonies that emerged on the plates were counted.

## RNA extraction and quantitative real-time PCR

The TRIzol Reagent (Invitrogen) was utilized for total RNA extraction, and subsequent cDNA synthesis was performed using the FastQuant RT Kit (Tiangen), which includes DNase treatment to eradicate genomic contamination, following the manufacturer's protocols. The expression profiles of individual genes were evaluated using the SYBR Premix Ex Taq™ (Takara), according to established procedures. Quantitative real-time PCR was carried out using an Applied Biosystems® QuantStudio 3 (Thermo Fisher Scientific). β-actin served as the internal control for mRNA. The primer sequences are detailed in Table EV1.

## Western blotting

Cellular lysates were prepared using 1× SDS-PAGE loading buffer. Protein extraction from cells was conducted, and quantification was performed using the BCA Protein Assay kit (Beyotime). Subsequently, the proteins were separated by SDS-PAGE (10%) and transferred onto PVDF membranes using a semidry blotting technique (Bio-Rad Trans Blot Turbo System). The membranes were then blocked with 5% BSA and probed with various antibodies. The TRAF6 antibody was diluted at 1:500 (Boster), the Ago2 antibody was diluted at 1:500 (Boster), and the anti-Flag, anti-Myc, and anti-Tubulin monoclonal antibodies were diluted at 1:1000 (Beyotime). Additionally, HRP-conjugated anti-rabbit IgG or anti-mouse IgG (Abbkine) was used at a dilution of 1:5000. The results were representative of three independent experiments. The immunoreactive proteins were visualized using WesternBrightTM ECL (Advansta), and digital imaging was conducted with a cold CCD camera.

## Co-immunoprecipitation

The cells were lysed in the cold IP buffer. The supernatant from cell lysates was incubated with indicated antibodies. Then, the protein A/G beads were incubated with the lysates overnight at 4 °C. The beads were washed three times with cold lysis buffer and resuspended in the protein loading buffer. The immunoprecipitates and whole-cell lysates (WCL) were followed by western blotting analysis.

## Immunofluorescence

Cells cultured on glass coverslips in 24-well plates were fixed with 4% paraformaldehyde for 15 min. following a triple wash with PBS. Subsequently, the cells were permeabilized with 0.5% TritonX-100 in PBS (PBST) for 15 min. After a 60-min. block with 5% BSA, the cells were subjected to a 10-h incubation with the primary antibody at 4 °C. Following three washes with PBS, the cells were exposed to a fluorescent-dye conjugated secondary antibody for 60 min. at room temperature. Post a 10-min. DAPI counterstain, the fluorescence signals of the cells were captured using a Leica fluorescence microscope.

## Dual-luciferase reporter assays

At 48 h after transfection, cell lysis for reporter activity was carried out using the Dual-Luciferase reporter assay system (Promega). Each construct was transfected in triplicate for each assay. Ratios of Renilla luciferase readings to firefly luciferase readings were determined for each experiment, and the averages of triplicates were calculated.

## Chromatin immunoprecipitation (ChIP) assay

MKC cells were inoculated in 10 cm$^2$ dishes and cells were transfected with plasmids after reaching $5.0 \times 10^6$, after which ChIP was performed. Briefly, cells were treated with 1% formaldehyde for 10 min at 37 °C, followed by Glycine Solution for 5 min at 25 °C to terminate the cross-linking. The cells were washed twice in PBS, lysed on ice, collected, and ultrasonically fragmented. Centrifugation at 4 °C,

12,000 rpm for 10 min. The appropriate antibody was added to the supernatant and rotated for 12 h at 4 °C. The Agarose Protein A/G beads were added to it, at 4 °C for 2 h. Centrifuge at 6000 rpm for 5 min and discard the supernatant. Washed the beads in sequence with low salt wash buffer, low salt wash buffer, LiCl wash buffer, and TE buffer. Finally, the beads were eluted with elution buffer and diluted to 500 ml with diffusion buffer. Added 5 M NaCl at 65 °C for 12 h, 0.5 M EDTA, 1 M Tris-HCl, and 20 mg/ml Proteinase K treated for 2 h at 45 °C.

## RNA binding protein immunoprecipitation (RIP)

For RNA immunoprecipitation (RIP) assays, MKC ($\sim 1.0 \times 10^7$) cells were harvested after 48 h transfection, and RIP assays were carried out with Magna RIP RNA-Binding Protein Immunoprecipitation Kit (Millipore) and anti-Flag or IgG antibody (Abcam) following the manufacturer's protocol. The MS2-RIP assay was also conducted in MKC cells ($\sim 1.0 \times 10^7$) transfected with pcDNA3.1-MS2, pcDNA3.1-MS2-7SL, pcDNA3.1-MS2-7SL-mut, or pMS2-GFP (Addgene). To construct plasmids that could produce 7SL identified by the MS2 protein, an MS2-12X fragment was cloned into these plasmids. In addition, a plasmid was constructed to express a fusion protein of GFP and MS2 gene, enabling the binding of the GFP-MS2 fusion protein to the MS2 fragment, which was subsequently identified using an anti-GFP antibody (Beyotime). Subsequently, RNA was extracted from the residual beads, and qPCR was utilized to assess the expression levels of miRNAs.

## DNA pull-down assay

Two primer pairs containing the predicted ADF-1L binding motifs (7SL-motif1 and 7SL-motif2) were synthesized in Jinweizhi Company (Suzhou Jinweizhi Biotechnology Co., Ltd., Suzhou, China), then biotin-labeled and unlabeled DNA probes were generated by PCR. The whole-cell lysates from MKC cells ($\sim 5.0 \times 10^6$) transfected with the ADF-1L expression plasmids were incubated with purified biotinylated DNA probes for 2 h at 25 °C. The complexes were isolated by streptavidin agarose beads (Invitrogen), followed by subsequent SDS-PAGE analysis and immunoblotting with anti-Flag antibody (Beyotime).

## EMSA assay

Recombinant ADF-1L proteins were mixed with biotin-labeled probes and competitor probes (unlabeled cold probes) in a reaction mixture (10 µl: 20 mM Tris-HCl pH 8.0, 1.5 mM MgCl$_2$, 1.5 mM DTT). After incubation at room temperature for 15 min, the reaction mixtures were conducted on 6% non-denaturing polyacrylamide gels and the EMSA was performed using Chemiluminescent EMSA Kit (Beyotime).

## RNA pull-down assay

To detect the binding of 7SL and Ago2, MKC cells ($\sim 1.0 \times 10^7$) were incubated with purified biotinylated 7SL for 2 h at 25 °C, followed by subsequent SDS-PAGE analysis and immunoblotting with anti-Ago2 antibody (Boster). To detect the binding site between 7SL and miR-2187-3p, 7SL and 7SL-mut with miR-2187-3p binding sites mutated were biotin-labeled with the T7 RNA polymerase and Biotin RNA

Labeling Mix (RiboBio), treated with RNase-free DNase I, and purified with RNeasy Mini Kit (Qiagen). The whole-cell lysates from MKC cells ($\sim 1.0 \times 10^7$) were incubated with two purified biotinylated transcripts for 2 h at 25 °C, followed by subsequent qPCR analysis to evaluate the expression level of miR-2187-3p.

## Statistical analysis

The data were presented as the mean ± standard error (SE) from a minimum of three independent triplicated experiments. A two-sided Student's *t* test was employed for data assessment. The relative gene expression data was obtained using the $2^{-\Delta\Delta CT}$ method, and inter-group comparisons were analyzed through one-way analysis of variance (ANOVA) followed by Duncan's multiple comparison tests. A significance level of $P < 0.05$ was considered statistically significant.

# Data availability

This study includes no data deposited in external repositories.

The source data of this paper are collected in the following database record: biostudies:S-SCDT-10_1038-S44319-025-00379-8.

# Peer review information

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

## Acknowledgements

This study was supported by the National Natural Science Foundation of China (31822057).

## Author contributions

**Shang Geng**: Conceptualization; Data curation; Validation; Investigation; Writing—original draft; Writing—review and editing. **Xing Lv**: Supervision;

Validation; Investigation. **Tianjun Xu**: Conceptualization; Supervision; Funding acquisition; Project administration; Writing—review and editing.

Source data underlying figure panels in this paper may have individual authorship assigned. Where available, figure panel/source data authorship is listed in the following database record: biostudies:S-SCDT-10_1038-S44319-025-00379-8.

## Disclosure and competing interests statement

The authors declare no competing interests.

# Expanded View Figures

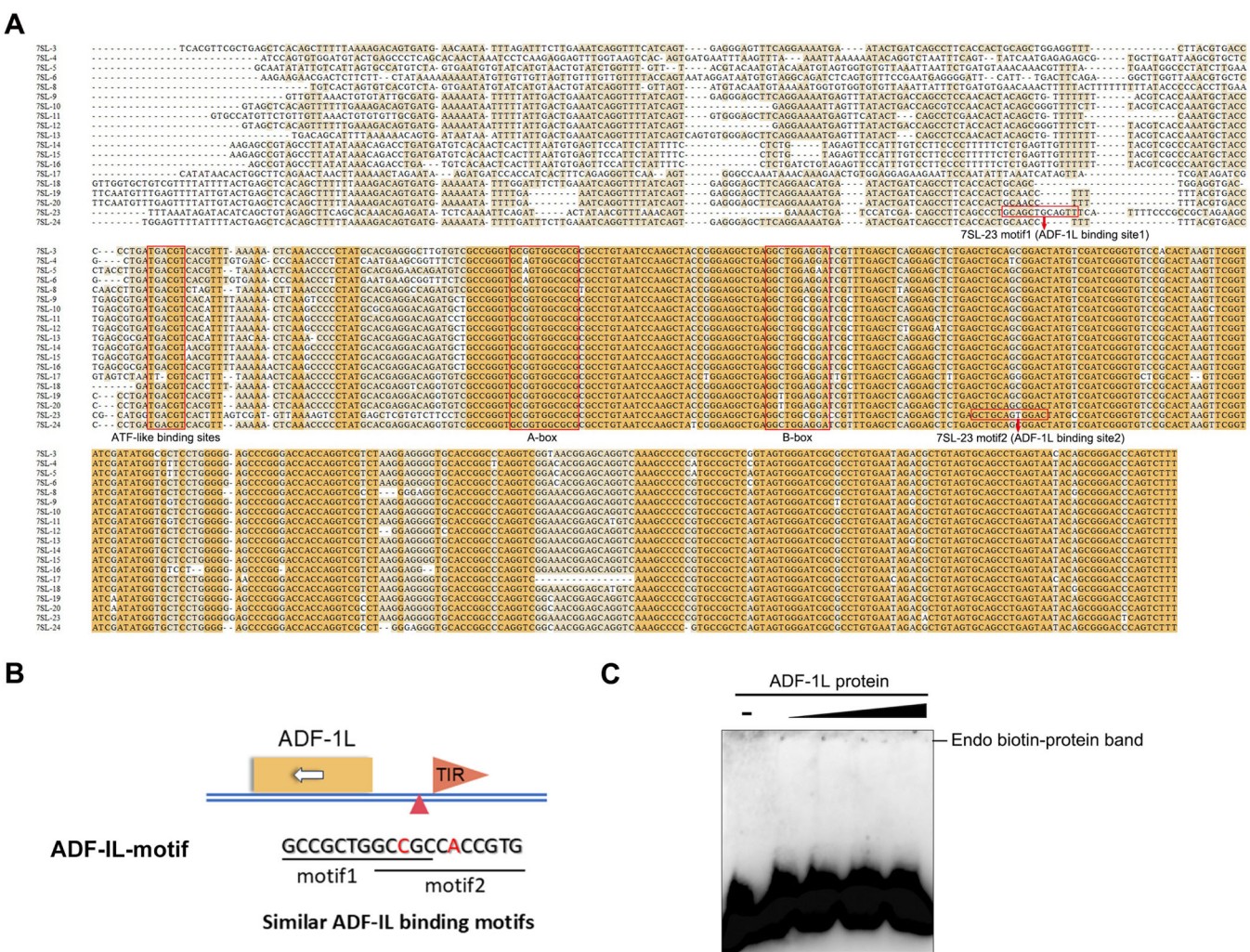

**Figure EV1.  Presentation of ADF-IL cis-binding sites.**

(A) The sequence alignment of 19 7SL gene with A box and B box (internal sequence + 200 bp external sequence), and the specific motif bound by ADF-IL has been labeled in 7SL-23. (B) A sequence of ADF-IL promoter containing two similar ADF-1L binding motifs. (C) EMSA assay performed with increasing concentrations of ADF-1L protein, and biotin-labeled probes (similar ADF-1L-motif). Source data are available online for this figure.

