## [Peer Review File · EMBO Reports]

PIF/harbinger transposon-derived protein serves as a cis-acting factor to promote 7SL-23 expression

Shang Geng, Xing Lv, and Tianjun Xu

Corresponding author(s): Tianjun Xu (tj-xu@shou.edu.cn)

Review Timeline:

Submission Date:	29th Mar 24
Editorial Decision:	9th Apr 24
Appeal Received:	24th Aug 24
Editorial Decision:	24th Oct 24
Revision Received:	22nd Nov 24
Editorial Decision:	13th Jan 25
Revision Received:	14th Jan 25
Accepted:	17th Jan 25

Editor: Esther Schnapp

Transaction Report:

9th Apr 2024
Dear Prof. Xu,

Thank you for the submission of your manuscript to EMBO reports. I have now read and discussed it with my colleagues here, and I am sorry to say that we all agree that it is not well suited for us.

We note that your study identifies ADF-1L in the croaker fish *Miichthys miiuy*. You show that ADF-1L is a protein derived from the Myb-protein of the PIF/harbinger transposon located upstream of the 7SL-23 gene, that it binds to and promotes the expression of its neighboring gene 7SL-23, that it boosts the expression of IFN-1 and TNF- α , that 7SL-23 knockdown reduces the increased expression of IFN-1 and TNF- α in ADF-1L-overexpressing MKC cells, that ADF-1L inhibits viral replication and bacterial invasion and that 7SL depletion reduces these effects, that 7SL acts as a miR-2187-3p sponge, and that 7SL counteracts the effect of miR-2187-3p on TRAF6 expression as well as the negative effect of miR-2187-3p on the luciferase activities of NF- κ B and IRF3 reporter genes. We think that this will be of interest to researchers in the field.

However, we also note that it was reported before that genes derived from PIF-transposons have been "domesticated" to new genes that contain MADF domains in *Drosophila*, as you mention. Also, the study is limited to *M. miiuy*, and the majority of the work is done in cultured *M. miiuy* kidney cells. For EMBO reports, stronger in vivo evidence would be required. We think that the manuscript does not provide a sufficient advance for consideration for publication here, and we have therefore decided not to proceed with in-depth review.

That said, we think that your work is an excellent candidate for our partner journal Life Science Alliance (<http://www.life-science-alliance.org/>; our broad scope Open Access journal published in partnership between the EMBO-, Rockefeller University-, and Cold Spring Harbor Laboratory Presses). Eric Sawey, Executive Editor of Life Science Alliance (e.sawey@life-science-alliance.org) would be pleased to send your manuscript for in-depth peer review; no reformatting is required. We very much hope that you will be interested in this option: please follow the link below for transfer.

For EMBO reports, I am sorry that I cannot be more positive this time, and I thank you once more for your interest in our journal.

Yours sincerely,

** As a service to authors, EMBO Press provides authors with the ability to transfer a manuscript that one journal cannot offer to publish to another journal, without the author having to upload the manuscript data again. To transfer your manuscript to another EMBO Press journal using this service, please click on Link Not Available

08/23, 2024

Editor of EMBO REPORTS

Shanghai, China

Dear Dr. Esther Schnapp,

We would like to **re-submit** our revised manuscript titled '*PIF/harbinger transposon-derived protein serves as a cis-acting factor for 7SL*' to EMBO REPORTS as "**Exploratory Report**" type for publication again after carefully revising the manuscript and inclusion of key experimental data elucidating the underlying mechanism.

Your comments in an earlier submission version (EMBOR-2024-59301V1):

We note that your study identifies ADF-1L in the croaker fish *Miichthys miiuy*. You show that ADF-1L is a protein derived from the Myb-protein of the PIF/harbinger transposon located upstream of the 7SL-23 gene, that it binds to and promotes the expression of its neighboring gene 7SL-23, that it boosts the expression of IFN-1 and TNF- α , that 7SL-23 knockdown reduces the increased expression of IFN-1 and TNF- α in ADF-1L-overexpressing MKC cells, that ADF-1L inhibits viral replication and bacterial invasion and that 7SL depletion reduces these effects, that 7SL acts as a miR-2187-3p sponge, and that 7SL counteracts the effect of miR-2187-3p on TRAF6 expression as well as the negative effect of miR-2187-3p on the luciferase activities of NF- κ B and IRF3 reporter genes. We think that this will be of interest to researchers in the field.

However, we also note that it was reported before that genes derived from PIF-transposons have been "domesticated" to new genes that contain MADF domains in *Drosophila*, as you mention. Also, the study is limited to *M. miiuy*, and the majority of the work is done in cultured *M. miiuy* kidney cells. For EMBO reports, stronger in vivo evidence would be required. We think that the manuscript does not provide a sufficient advance for consideration for publication here, and we have therefore decided not to proceed with in-depth review.

Response to your comments:

Your attention to our manuscript seems to mainly focus on the immune function of ADF-1L and 7SL; However, the primary innovation of this manuscript lies in the introduction of the term "**cis-acting factor**". We recognize that our initial submission may not have effectively conveyed this innovation. Consequently, we have revised the abstract, introduction, results and discussion sections to better highlight this key point. Furthermore, to more explicitly illustrate the conceptual

findings of this study, we reorganized the figure layout and combined Figures 3-5 into a single Figure 3, which depicts the biological significance of ADF-1L cis-regulation of 7SL-23. We also added new data of Figure 2G-2Q to clarify the specific regulatory mechanism of ADF-1L.

Transcriptional regulation is a fundamental mechanism that influences gene expression, primarily governed by "cis-acting DNA elements" and "trans-acting protein factors." **This study challenges the traditional perspective that attributes cis-regulation exclusively to DNA elements.** Our findings suggest that certain proteins can exert cis-regulatory effects on adjacent genes within the chromosome, thus introducing the term "cis-acting factor." We posit that this concept represents a significant advancement in our understanding of transcriptional regulation. To validate the advancement of this concept and avoid superficial claims, we undertook the following investigations: First, our data indicate that the ADF-1L protein is derived from the PIF/harbinger transposon, suggesting that its cis-regulatory function may stem from the inherent self-preferencing of proteins associated with this transposon. Second, we provide evidence that ADF-1L can bind to and regulate the adjacent 7SL-23 gene, aligning with the definition of cis-regulation. Third, our revised manuscript includes additional findings elucidating the specific regulatory mechanisms of ADF-1L. In light of these points, we contend that the introduction of the new concept of "cis-acting factor" is credible, fulfilling the journal's criteria. However, acknowledging your concerns regarding the strength of our experimental data, we intend to submit our revised manuscript to Exploratory Report, a short scholarly research papers that present provocative scientific hypotheses based on initial data. As you mentioned, our research is restricted to *M. miiuy*, and we anticipate that future studies will identify additional proteins that function as cis-acting factors to further enrich our proposed concept.

In addition, we realize that in previous versions, our case study may have been too preliminary to guarantee the review and evaluation of the importance of proposed regulatory circuit. Consequently, as mentioned above, we have added key experimental data in the revised manuscript to clarify the host transcription mechanism utilized by ADF-1L, enhancing the comprehensiveness of our research and the clarity of the mechanism. Specifically, our current research illustrates that the **ADF-1L protein, originating from the PIF/harbinger transposon, selectively recruits histone acetyltransferase KAT2B in a MADF domain-dependent manner, thereby facilitating its nuclear translocation** and cis-regulating 7SL-23 (Fig. 2). From an

evolutionary perspective, given the high conservation of KAT from yeast to mammals and the widespread distribution of MADF gene family in non-mammalian organisms, **our study also offers direct and valuable insights into the functionality of the MADF gene family**. In light of these improvements, we kindly ask for your consideration in re-evaluating our manuscript.

To facilitate your evaluation and review, we have prepared summaries of the two versions for comparative analysis:

The abstract of our earlier submission version (EMBOR-2024-59301V1):

Transcriptional regulation is a critical mechanism that governs gene expression levels, primarily controlled by cis-acting DNA elements and trans-acting protein factors. DNA elements play an indispensable role in the cis-regulation of adjacent genes on chromosomes, while the regulatory molecules for cis-acting effects are not confined to DNA sequences. Our research indicates that transposon-derived proteins may retain their original DNA-binding preference and exert cis-regulatory effects on nearby genes on the chromosome, thus denoted as "cis-acting factors". In this study, ADF-1L, a protein derived from the Myb-protein of the PIF/harbinger transposon, has been found to bind to and exert cis-regulation on its neighboring gene, 7SL-23. Furthermore, ADF-1L protein boosts the host's resistance to pathogens by promoting the expression of immune molecule 7SL RNA. In summary, our findings expand the types of molecules that can exert cis-function in gene regulation and underscore the significance of transposons-derived sequences in the cellular processes and the evolutionary trajectory of the biological genome.

The abstract of our revised manuscript:

Transcriptional regulation is a critical mechanism that governs gene expression levels, primarily controlled by "cis-acting DNA elements" and "trans-acting protein factors". **However, the conventional view that cis-regulation is solely attributable to DNA elements has been challenged in this study**. Our research indicates that transposon-derived proteins may retain their original DNA-binding preference and exert cis-regulatory effects on nearby genes on the chromosome, thus denoted as "cis-acting factors". Specifically, our research demonstrates that the ADF-1L protein, derived from the PIF/harbinger transposon, **recruits the histone**

acetyltransferase KAT2B in a MADF domain-dependent manner, facilitating its nuclear translocation and binding to and cis-regulating its adjacent gene 7SL-23. Subsequently, ADF-1L protein boosts the host's resistance to pathogens by promoting the expression of immune molecule 7SL RNA. In summary, our findings expand the types of molecules that can exert cis-function in gene regulation and underscore the significance of transposons-derived sequences in the cellular processes and the evolutionary trajectory of the biological genome.

All authors have seen the manuscript and approved to submit to your journal. Neither the entire paper nor any part of its content has been published or has been accepted elsewhere. It is not being submitted to any other journal. Thanks very much for your attention and consideration.

Best regards.

Yours sincerely,

Dr. Tianjun Xu

Dear Prof. Xu,

Thank you for the submission of your manuscript to EMBO reports. We have now received the full set of referee reports that is pasted below.

As you will see, the referees acknowledge that the findings are potentially interesting. However, they also have several suggestions for how the study should be strengthened and I think all suggestions are good and should be addressed. Please let me know in case you disagree and we can discuss the exact revision requirements further, also in a video chat, if you like.

I would thus like to invite you to revise your manuscript with the understanding that the referee concerns must be fully addressed and their suggestions taken on board. Please address all referee concerns in a complete point-by-point response. Acceptance of the manuscript will depend on a positive outcome of a second round of review. It is EMBO reports policy to allow a single round of major revision only and acceptance or rejection of the manuscript will therefore depend on the completeness of your responses included in the next, final version of the manuscript.

We realize that it is difficult to revise to a specific deadline. In the interest of protecting the conceptual advance provided by the work, we recommend a revision within 3 months (24th Jan 2025). Please discuss the revision progress ahead of this time with the editor if you require more time to complete the revisions.

- 1) A data availability section providing access to data deposited in public databases is missing. If you have not deposited any data, please add a sentence to the data availability section that explains that.
- 2) Your manuscript contains statistics and error bars based on $n=2$. Please use scatter blots in these cases. No statistics should be calculated if $n=2$.

5) a complete author checklist, which you can download from our author guidelines . Please insert information in the checklist that is also reflected in the manuscript. The completed author checklist will also be part of the RPF.

6) Please note that all corresponding authors are required to supply an ORCID ID for their name upon submission of a revised

manuscript (). Please find instructions on how to link your ORCID ID to your account in our manuscript tracking system in our Author guidelines

- the name of the statistical test used to generate error bars and P values,
- the number (n) of independent experiments (please specify technical or biological replicates) underlying each data point,
- the nature of the bars and error bars (s.d., s.e.m.),
- If the data are obtained from n {less than or equal to} 2, use scatter blots showing the individual data points.

12) All Materials and Methods need to be described in the main text using our 'Structured Methods' format, which is required for all research articles. According to this format, the Methods section includes a separate file called Reagents and Tools Table (listing key reagents, experimental models, software and relevant equipment and including their sources and relevant identifiers) followed by a Methods and Protocols section describing the methods using a step-by-step protocol format. The aim is to facilitate adoption of the methodologies across labs. More information on how to adhere to this format as well as a downloadable template (.docx) for the Reagents and Tools Table can be found in our author guidelines: <https://www.embopress.org/page/journal/14693178/authorguide#structuredmethods>.

An example of a Method paper with Structured Methods can be found here: <https://www.embopress.org/doi/full/10.1038/s44320-024-00037-6#sec-4>.

As part of the EMBO publication's Transparent Editorial Process, EMBO reports publishes online a Review Process File (RPF) to accompany accepted manuscripts. This File will be published in conjunction with your paper and will include the referee

reports, your point-by-point response and all pertinent correspondence relating to the manuscript.

I look forward to seeing a revised form of your manuscript when it is ready.

Yours sincerely,

Referee #1:

This is a nice paper reporting on the transcriptional activities of the ADF-1L domesticated transposon-encoded protein. Authors demonstrate that ADF-1L regulates an antimicrobial defence reaction through modulating 7SL transcript levels. Altogether the experiments are well executed and the conclusions are justified. Nonetheless, I list a couple of points that authors should address in a revision:

- 1) What remained somewhat ambiguous, if the 7SL-23 copy is the only target of ADF-1L? Given the fact that one of the binding sites of ADF-1L is inside the 7SL-23 gene (shown in Fig. 2A), one could expect that such binding site would also be present in at least some of the other 23 7SL genes. A sequence alignment of the respective regions of all 24 7SL genes should be presented to provide the reader with evidence that interactions of ADF-1L are unique for the 7SL-23 gene.
- 2) Related to the point above, authors provide evidence of transcriptional regulation of the 7SL-23 gene by ADF-1L, but it remained ambiguous if transcript levels of the other 23 7SL genes are also regulated by ADF-1L? Please address.
- 3) Related to the above two points, if any 7SL gene other than 7SL-23 would also respond to ADF-1L regulation, that would weaken the cis-regulatory model proposed by authors. Please clarify this point.
- 4) Fig. 11 displays a model in which ADF-1L binds to the TIR region of the ADF-1L gene itself. This model is at odds with the model presented in Fig. 2A. Since the myb-like proteins encoded by Harbinger transposons have been shown to bind to several subterminal sequence motifs in the TIR region of the transposon, the model presented in Fig. 11 is a viable one. Authors should look for the presence of the ADF-1L binding motif within the TIR. If such motif can be found in the TIR, that would raise the possibility of transcriptional regulation of the ADF-1L gene itself and a positive feedback loop (binding of ADF-1L regulates itself). This needs to be addressed in a revision.
- 5) Authors should experimentally demonstrate locus-specific enrichment of histone acetylation at the 7SL-23 locus by overexpression of ADF-1L. This is predicted by the model they propose.

Minor:

- 1) On page 3, line 59, please explain what "losing transpositional ability" means. The general reader may not be familiar with the processes that lead to the so-called domesticated transposase-derived proteins.

Referee #2:

The manuscript investigates the regulation of 7SL RNA expression and its interaction with the transposon-derived protein ADF-1L, which acts as a cis-acting factor regulating its expression. The study identifies that ADF-1L recruits KAT2B histone acetyltransferase to enhance 7SL transcriptional activity and promoting nuclear translocation. Moreover, the study identified that ADF-1L overexpression enhances immune responses by increasing the levels of IFN-I and TNF- α during infections. Similarly, 7SL RNA was shown to play a role in immune signaling when its levels are altered through knockdown or overexpression approaches. The combined effects of ADF-1L and 7SL RNA were found to inhibit viral replication and bacterial infections. Additionally, 7SL RNA acts as a competing endogenous RNA by binding to the RNA-induced silencing complex (RISC) and interacting with specific miRNAs, particularly miR-2187-3p, further enhancing interferon response and immune regulation. Overall, the claims of the manuscript are well supported and controlled. Moreover, the findings are likely of being a broad interest.

Referee #3:

The authors present a model where transposon-derived ADF-1L, through interactions with KAT2B, serves to regulate expression of 7SL-23, a member of 7SL RNA family. Moreover, they show that 7SL-23 can be bound and regulated by microRNAs, and argue that 7SL-23 acts to inhibit microRNA-mediated downregulation of TRAF6, an immune regulator. The authors emphasize the genomic proximity of ADF-1L and 7SL-23 as an example of cis-regulation by ADF-1L of 7SL-23 expression. Although this is a key take-away message, there is little discussion of whether this proximity is indeed related to 7SL-23 regulation, and the manuscript would be strengthened with the following recommendations:

Figure 1

Although the proximity of the ADF-1 locus to the 7SL-23 locus is intriguing, it is unclear how this arrangement is mechanistically related to their regulatory relationship, and therefore, would limit its appeal to a broader audience.

- M. miiuy contains 24 7SL genes. What impact does infection have on transcription across all 24 copies? Include these data to highlight the significance of 7SL-23 upregulation specifically
- Authors consider ADF-1L as a 'cis-acting factor', in that it acts in cis to regulate 7SL-23, based on its genomic locus in close proximity to the 7SL-23 locus and changes in 7SL-23 expression in response to ADF-L manipulation. Are putative binding sequences present in other 7SL copies? More importantly, what impact does ADF-L KD/overexp have on expression of other 7SL copies?

Figure 2:

Although an interaction with KAT2B suggests a mechanism for transcriptional upregulation, it is unclear whether acetyltransferase activity is involved.

- The authors show that ADF-1L directly binds to sequences within the 7SL-23 gene. In addition, ADF-L directly binds to the histone acetylase KAT2B. The authors show KAT2B interactions at the 7SL-23 locus are ADF-1L dependent, suggesting that 7SL-23 upregulation by ADF-1L is mediated through the acetyltransferase activity of KAT2B. To confirm this, the authors should examine: 1) whether KAT2B-deltaHAT impacts 7SL-23 expression and 2) acetylation status at the 7SL-23 locus

Figure 3:

Although the reporter and ectopic expression data are intriguing, it is unclear whether microRNA regulation is biologically relevant within the context of physiological levels of 7SL-23 and respective target microRNAs. As such, this component of the manuscript is hard to interpret.

- The authors show that the 7SL-23 RNA contains microRNA binding sites for multiple miRNAs, one (miR-2187) of whose levels are impacted by 7SL KD/overexpression. Further, the authors show that miR-2187 target sites are present in TRAF mRNA, and through a series of overexpression expts, confirm that TRAF mRNA and downstream target expression levels can be modulated by miR-2187 and/or 7SL-23 overexpression. Although the reporter data are convincing, it is hard to assess whether the role of endogenous 7SL-23 is to serve as a 'sponge', given that these experiments rely on overexpression. To test the biological relevance of this model directly, the authors should assess whether 7SL-23 knockdown (as well as ADF-L knockdown) results in 1) increased miR-2187 expression and 2) decreased TRAF mRNA expression.

Statement for correction

Thank you for the insightful comments provided by the three referees and editor regarding our manuscript. Their feedback has been invaluable in guiding the revision process and enhancing the quality of our work, as well as offering significant direction for our research. We have carefully addressed all the comments and made the necessary revisions, which we hope will meet with your approval. Additionally, due to the further enrichment of the experimental data and the limitations of the three figures in the exploratory report, we have adjusted the paper format from an exploratory report to a short report, in accordance with the senior editor's suggestion. Below are the main corrections made to the paper, along with our responses to the referees' comments:

Referee #1:

This is a nice paper reporting on the transcriptional activities of the ADF-1L domesticated transposon-encoded protein. Authors demonstrate that ADF-1L regulates an antimicrobial defence reaction through modulating 7SL transcript levels. Altogether the experiments are well executed and the conclusions are justified. Nonetheless, I list a couple of points that authors should address in a revision:

Response: Many thanks for your appreciation and professional comments of this manuscript. We have carefully revised of the article according to your advice. Specific changes were listed below:

Q1. What remained somewhat ambiguous, if the 7SL-23 copy is the only target of ADF-1L? Given the fact that one of the binding sites of ADF-1L is inside the 7SL-23 gene (shown in Fig. 2A), one could expect that such binding site would also be present in at least some of the other 23 7SL genes. A sequence alignment of the respective regions of all 24 7SL genes should be presented to provide the reader with evidence that interactions of ADF-1L are unique for the 7SL-23 gene.

Response: Due to the presence of pseudogenes for 7SL, some of their sequences are incomplete and lack transcriptional activity. We collected 19 7SL gene sequences with more than 90% similarity to 7SL-23, all of which retain the A-box and B-box, based on the whole genome, and performed sequence alignment (Fig. EV2A). Notably, the two ADF-1L binding sites, motif1 and motif2, in 7SL-23 are unique to 7SL-23, even though motif2 is located within the internal sequence of 7SL-23 (Fig. EV2A). Additionally, we performed sequencing analysis on the precipitated products from the ChIP experiment involving ADF-1L, and the motif results specifically matched 7SL-23 (Fig. 2B). Therefore, ADF-1L specifically binds to 7SL-23, and this conclusion is further validated by additional experimental data in the following response.

Q2. Related to the point above, authors provide evidence of transcriptional regulation of the 7SL-23 gene by ADF-1L, but it remained ambiguous if transcript levels of the other 23 7SL genes are also regulated by ADF-1L? Please address.

Response: Due to the high sequence conservation within the internal regions of 7SL RNA, it is challenging to design specific RT primers for different 7SL genes to directly assess the regulatory effect of ADF-1L on the transcription levels of various 7SL RNAs. 7SL is transcribed by RNA polymerase III, and the transcription factor complex TFIIC is essential in this process. Therefore, we plan to conduct ChIP experiments to examine the binding affinity between TFIIC and the various 7SL genes to evaluate their transcriptional activity. We have constructed plasmids for the key subunits of the TFIIC complex, TFIIC63 and TFIIC90, and designed specific ChIP primers targeting the external promoter sequences of different 7SL genes (including 7SL-23 from this study, 7SL-17, which lacks transcriptional activity, and the actively transcribed 7SL-16 and 7SL-18) to assess the impact of ADF-1L on the transcription of these 7SL genes. Results showed that ADF-1L knockdown specifically reduces the expression of 7SL-23 without affecting 7SL-16 or 7SL-18, highlighting the selective regulation of 7SL-23 by ADF-1L (Fig. EV2A, Fig. 2H).

These findings firmly establish ADF-1L as a cis-regulatory factor that controls the transcriptional activity of the 7SL-23 gene.

Q3. Related to the above two points, if any 7SL gene other than 7SL-23 would also respond to ADF-1L regulation, that would weaken the cis-regulatory model proposed by authors. Please clarify this point.

Response: As shown above, we performed sequence alignment analysis and CHIP experiments, and the results indicate that ADF-1L specifically regulates 7SL-23, without affecting other 7SL genes (7SL-16 and 7SL-18) (Fig. EV2A, Fig. 2H). This supports the cis-regulatory model we proposed. Once again, thank you for your professional advice.

Q4. Fig. 1I displays a model in which ADF-1L binds to the TIR region of the ADF-1L gene itself. This model is at odds with the model presented in Fig. 2A. Since the myb-like proteins encoded by Harbinger transposons have been shown to bind to several subterminal sequence motifs in the TIR region of the transposon, the model presented in Fig. 1I is a viable one. Authors should look for the presence of the ADF-1L binding motif within the TIR. If such motif can be found in the TIR, that would raise the possibility of transcriptional regulation of the ADF-1L gene itself and a positive feedback loop (binding of ADF-1L regulates itself). This needs to be addressed in a revision.

Response: We greatly appreciate your insightful comments on the differences between our models. Investigating the self-regulation of ADF-1L is an excellent suggestion, and we are thankful for your practical experimental ideas and recommendations. According to your suggestion, We designed CHIP primers flanking the TIR sequence in the ADF-1L promoter region and found that ADF-1L indeed binds to its own promoter, suggesting the possibility of self-regulation (Fig. 2B). Furthermore, we discovered that ADF-1L can recruit KAT2B to bind its own promoter and promote acetylation modification at its gene locus and ADF-1L expression (Fig.

3C, 3D, 3H, 3M, and 3N). However, we did not find a perfect match for the ADF-1 binding motif in the ADF-1L promoter sequence, although there is a similar sequence (Fig. EV2B). We attempted EMSA experiments to further verify their binding but found no interaction (Fig. EV2C). Nevertheless, the additional experiments we provided can already support the notion that ADF-1L can achieve self-regulation, further reinforcing the model we proposed. Finally, we made slight modifications to the model presented in Fig. 1I, as it has been shown that the Myb-like protein encoded by the Harbinger transposon binds to internal transposon sequences near the TIR, rather than the TIR sequence itself (DOI10.1073/pnas.0707746105). Additionally, since ADF-1L is transcribed in the same direction as 7SL-23, it has the potential to bind both its own promoter and the promoter of the adjacent 7SL-23 gene. We have updated this in the Fig. 1I, making the model more consistent with our existing experimental data.

Q5. Authors should experimentally demonstrate locus-specific enrichment of histone acetylation at the 7SL-23 locus by overexpression of ADF-1L. This is predicted by the model they propose.

Response: Thank you for your professional advice. We have conducted additional experiments based on your suggestions. Results showed that acetylation modifications were indeed present at the 7SL-23 and ADF-1L gene loci, and it was found that ADF-1L, rather than ADF-1L- Δ MADF, can increase the acetylation levels at these loci (Fig. 3M and 3N). These results support the view that the MADF domain of ADF-IL plays a key role in the regulation of histone acetylation.

Minor:

Q1. On page 3, line 59, please explain what "losing transpositional ability" means. The general reader may not be familiar with the processes that lead to the so-called domesticated transposase-derived proteins.

Response: When a transposable element sequence undergoes a mutation, it may lose

the ability to transpose within the genome. Thank you for your suggestion; we made the necessary changes to improve the readability of the manuscript (lines 59-61).

Referee #2:

The manuscript investigates the regulation of 7SL RNA expression and its interaction with the transposon-derived protein ADF-1L, which acts as a cis-acting factor regulating its expression. The study identifies that ADF-1L recruits KAT2B histone acetyltransferase to enhance 7SL transcriptional activity and promoting nuclear translocation. Moreover, the study identified that ADF-1L overexpression enhances immune responses by increasing the levels of IFN-I and TNF- α during infections. Similarly, 7SLRNA was shown to play a role in immune signaling when its levels are altered through knockdown or overexpression approaches. The combined effects of ADF-1L and 7SL RNA were found to inhibit viral replication and bacterial infections. Additionally, 7SL RNA acts as a competing endogenous RNA by binding to the RNA-induced silencing complex (RISC) and interacting with specific miRNAs, particularly miR-2187-3p, further enhancing interferon response and immune regulation. Overall, the claims of the manuscript are well supported and controlled. Moreover, the findings are likely of being a broad interest.

Response: We sincerely appreciate your positive feedback on the manuscript. We are glad to know that the claims of our study, especially regarding the regulation of 7SL RNA expression and its interaction with the ADF-1L protein, were clearly presented and well-supported. Your recognition of the broad interest of these findings further motivates us to continue exploring this area of research.

Referee #3:

The authors present a model where transposon-derived ADF-1L, through interactions with KAT2B, serves to regulate expression of 7SL-23, a member of 7SL RNA family.

Moreover, they show that 7SL-23 can be bound and regulated by microRNAs, and argue that 7SL-23 acts to inhibit microRNA-mediated downregulation of TRAF6, an immune regulator. The authors emphasize the genomic proximity of ADF-1L and 7SL-23 as an example of cis-regulation by ADF-1L of 7SL-23 expression. Although this is a key take-away message, there is little discussion of whether this proximity is indeed related to 7SL-23 regulation, and the manuscript would be strengthened with the following recommendations:

Response: Thank you for your thoughtful and constructive feedback. In response to your and Referee 1's recommendation, we have expanded the manuscript to include a more detailed discussion on how this genomic proximity between ADF-1L and 7SL-23 may influence 7SL-23 regulation. We believe this addition strengthens our argument for the cis-regulatory role of ADF-1L in 7SL-23 expression. Thank you again for your insightful comments and suggestions. Specific changes were listed below.

Figure 1

Q: Although the proximity of the ADF-1 locus to the 7SL-23 locus is intriguing, it is unclear how this arrangement is mechanistically related to their regulatory relationship, and therefore, would limit its appeal to a broader audience.

- *M. miiuy* contains 24 7SL genes. What impact does infection have on transcription across all 24 copies? Include these data to highlight the significance of 7SL-23 upregulation specifically.

Response: Based on your suggestion, we examined the expression of 24 7SL genes using whole-genome and transcriptome data in uninfected, SCRV-infected, and *Vibrio anguillarum*-infected conditions (Fig. EV1). Some 7SL genes have lost their transcriptional activity, while the active 7SL genes exhibited different responses to pathogen infections. Some were downregulated after infection (7SL-18, 7SL-19, 7SL-20, and 7SL-24), while others were specifically upregulated in response to pathogenic infections, similar to 7SL-23, which was also upregulated post-infection

(7SL-9 to 7SL-16). However, it is noteworthy that the read depth of 7SL-23 is much higher than that of other 7SL genes, suggesting its highest transcriptional activity (Fig. 1B and Fig. EV1). ChIP experiments further confirmed this by examining the binding affinity between the TFIIC transcription factor complex and different 7SL genes (Fig. 2G). This also indicates the unique biological significance of ADF-1L's regulation of 7SL-23.

Q: - Authors consider ADF-1L as a 'cis-acting factor', in that it acts in cis to regulate 7SL-23, based on its genomic locus in close proximity to the 7SL-23 locus and changes in 7SL-23 expression in response to ADF-L manipulation. Are putative binding sequences present in other 7SL copies? More importantly, what impact does ADF-L KD/overexp have on expression of other 7SL copies?

Response: Due to the presence of pseudogenes for 7SL, some of their sequences are incomplete and lack transcriptional activity. We collected 19 7SL gene sequences with more than 90% similarity to 7SL-23, all of which retain the A-box and B-box, based on the whole genome, and performed sequence alignment (Fig. EV2A). Notably, the two ADF-1L binding sites, motif1 and motif2, in 7SL-23 are unique to 7SL-23 (Fig. EV2A).

Due to the high sequence conservation within the internal regions of 7SL RNA, it is challenging to design specific RT primers for different 7SL genes to directly assess the regulatory effect of ADF-1L on the transcription levels of various 7SL RNAs. 7SL is transcribed by RNA polymerase III, and the transcription factor complex TFIIC is essential in this process. Therefore, we plan to conduct ChIP experiments to examine the binding affinity between TFIIC and the various 7SL genes to evaluate their transcriptional activity. We have constructed plasmids for the key subunits of the TFIIC complex, TFIIC63 and TFIIC90, and designed specific ChIP primers targeting the external promoter sequences of different 7SL genes (including 7SL-23 from this study, 7SL-17, which lacks transcriptional activity, and the actively transcribed 7SL-16 and 7SL-18) to assess the impact of ADF-1L on the

transcription of these 7SL genes. Results showed that ADF-1L knockdown specifically reduces the expression of 7SL-23 without affecting 7SL-16 or 7SL-18, highlighting the selective regulation of 7SL-23 by ADF-1L (Fig. 2G and 2H). These findings firmly establish ADF-1L as a cis-regulatory factor that controls the transcriptional activity of the 7SL-23 gene.

Finally, as suggested by Referee 1, the self-regulation of ADF-IL offers valuable insights for investigating the regulatory relationship arising from the genomic proximity between ADF-IL and 7SL-23. Our study demonstrates that ADF-IL can recruit KAT2B, bind to its own promoter, alter the acetylation modification levels at the promoter site, and influence its own expression (Fig. 3C, 3D, 3H, 3M, and 3N). This indicates that ADF-IL can not only cis regulate 7SL-23, but also cis regulate its own expression. Furthermore, their consistent transcriptional direction suggests that their genomic arrangement indeed impacts their regulatory relationship (Fig. 2I).

Figure 2:

Q: Although an interaction with KAT2B suggests a mechanism for transcriptional upregulation, it is unclear whether acetyltransferase activity is involved.

- The authors show that ADF-1L directly binds to sequences within the 7SL-23 gene. In addition, ADF-1L directly binds to the histone acetylase KAT2B. The authors show KAT2B interactions at the 7SL-23 locus are ADF-1L dependent, suggesting that 7SL-23 upregulation by ADF-1L is mediated through the acetyltransferase activity of KAT2B. To confirm this, the authors should examine: 1) whether KAT2B- Δ HAT impacts 7SL-23 expression and 2) acetylation status at the 7SL-23 locus

Response: As suggested by Referee 1, our supplementary experiments have confirmed that ADF-IL can also achieve self-cis-regulation (Fig. 2B, 3C, 3D). Therefore, when examining the involvement of KAT2B acetyltransferase activity and the acetylation status of relevant sites based on your recommendation, we not only analyzed 7SL-23 but also additionally examined the ADF-IL gene.

1) Impact of KAT2B- Δ HAT on 7SL-23 and ADF-IL expression: KAT2B can

promote the expression of 7SL-23 and ADF-IL, but KAT2B- Δ HAT cannot (Fig.3H). This indicates that KAT2B indeed regulates their expression through its acetyltransferase activity.

2) Acetylation status at the 7SL-23 and ADF-IL locus: Acetylation modifications were indeed present at the 7SL-23 and ADF-1L gene loci, and it was found that ADF-1L, rather than ADF-1L- Δ MADF, can increase the acetylation levels at these loci (Fig. 3M and 3N). These findings further support the role of ADF-1L in mediating transcriptional upregulation via KAT2B acetyltransferase activity.

Figure 3:

Q: Although the reporter and ectopic expression data are intriguing, it is unclear whether microRNA regulation is biologically relevant within the context of physiological levels of 7SL-23 and respective target microRNAs. As such, this component of the manuscript is hard to interpret.

- The authors show that the 7SL-23 RNA contains microRNA binding sites for multiple miRNAs, one (miR-2187) of whose levels are impacted by 7SL KD/overexpression. Further, the authors show that miR-2187 target sites are present in TRAF mRNA, and through a series of overexpression expts, confirm that TRAF mRNA and downstream target expression levels can be modulated by miR-2187 and/or 7SL-23 overexpression. Although the reporter data are convincing, it is hard to assess whether the role of endogenous 7SL-23 is to serve as a 'sponge', given that these experiments rely on overexpression. To test the biological relevance of this model directly, the authors should assess whether 7SL-23 knockdown (as well as ADF-L knockdown) results in 1) increased miR-2187 expression and 2) decreased TRAF mRNA expression.

Response: Thank you for your valuable suggestions. To directly test the biological relevance of the proposed model, we assessed the mRNA expression levels of miR-2187-3p and TRAF6 following knockdown of 7SL-23 and ADF-IL. Our results demonstrated that knockdown of 7SL-23 and ADF-IL led to increased miR-2187-3p

expression and decreased TRAF6 expression (Fig. 5D, 5N, 5P, and 5R). Additionally, we observed that knockdown of ADF-IL rescued the miR-2187-3p downregulation and TRAF6 upregulation induced by 7SL-23 overexpression (Fig. 5Q and 5S). These findings further highlight the regulatory role of ADF-IL within the 7SL/miR-2187-3p/TRAF6 axis.

Dear Prof. Xu,

Thank you for your patience while your revised manuscript was re-reviewed. We have now received the enclosed reports from referees 1 and 3, who were asked to assess it. Referee 3 still has a few more comments and suggestions that I would like you to address and incorporate before we can proceed with the official acceptance of your manuscript. Please co-submit a detailed point-by-point response with your final ms.

A few editorial requests will also need to be addressed:

- Please correct the conflict of interest subheading to "Disclosure Statement and Competing Interests"
- The corresponding author needs to have an institutional email address
- The author credits need to be removed from the ms file. All credits are entered during online ms submission.
- Materials and methods should be called "Methods"

Figure Legends - Comments

- Please note that the legend for figure 4k is missing in the manuscript. This needs to be rectified.
- Please note that the exact p values are not provided in the legends of figures 1d-e, g, k-m; 2g-h; 3e, h, j, l, n; 4a-g, i; 5b, d-j, l-s.
- Please note that in figures 1d-e, g, k-m; 2g-h; 4a-g, i; there is a mismatch between the annotated p values in the figure legend and the annotated p values in the figure file that should be corrected.

I would like to suggest some changes to the abstract that needs to be written in present tense. Please let me know whether you agree with these changes and please also modify the abstract in response to referee 3's comments, if necessary:

Transcriptional regulation governs gene expression levels, primarily controlled by "cis-acting DNA elements" and "trans-acting protein factors". However, the conventional view that cis-regulation is solely attributable to DNA elements is challenged in this study. Our research indicates that transposon-derived proteins may retain their original DNA-binding preference and exert cis-regulatory effects on nearby genes on the chromosome, thus denoted as "cis-acting factors". Specifically, we show that the ADF-1L protein, derived from the PIF/harbinger transposon, recruits the histone acetyltransferase KAT2B in a MADF domain-dependent manner, facilitating its own nuclear translocation and binding to and cis-regulating its own and adjacent gene 7SL-23. ADF-1L protein also boosts the host's resistance to pathogens by promoting the expression of immune molecule 7SL RNA. In summary, our findings expand the types of molecules that can exert cis-function in gene regulation and underscore the relevance of transposons-derived sequences in the cellular processes.

Referee #1:

Authors have addressed my concerns in a satisfactory manner in this revision.

Referee #3:

In their response, the authors state that 'active 7SL genes exhibited different responses to pathogen infections'. These data should be presented in Figure 1, in order to assess/compare to the response of 7SL-23, as well as to CHIP data shown in Figure 2G. Varied responses among more distant copies weakens, rather than strengthens, the argument that 7SL-23 regulation by ADF-1L is based on its physical proximity, a major claim of manuscript (e.g. 'cis-acting' nature highlighted in manuscript).

In their response, the authors state that, 'Results showed that ADF-1L knockdown specifically reduces the expression of 7SL-23

without affecting 7SL-16 or 7SL-18, highlighting the selective regulation of 7SL-23 by ADF-1L (Fig. 2G and 2H)'. Rather than infer from CHIP data, the authors should measure the effect of ADF-1L KD on the expression of active 7SL genes (e.g. 7SL-16 and 7SL-18) to show that ADF1L activity is specific to 7SL-23, and, as authors claim in response, not 7SL-16 or 7SL-18.

Overall, the inclusion of additional data has strengthened the authors' claims. That said, it remains unclear whether the physical proximity is mechanistically-related to the observed regulation of 7SL-23 by ADF1L. Ectopic expression of ADF-1L also leads to the upregulation of 7SL-23. Given that the 'cis-acting' nature of ADF1-L is central to the manuscript's narrative, more clarification of the novel nature would be warranted. For example, are all transcription factors that regulate their own expression by binding to their own promoters considered 'cis-acting factors'?

Statement for correction

Thank you for the valuable comments provided by the editor and referee #3 regarding our manuscript. We highly respect their feedback, which has been instrumental in guiding the revision process and significantly improving the quality of our work. We have carefully reviewed each suggestion and fully incorporated the requested revisions. We believe these changes have strengthened our manuscript, and we hope they meet your approval. Below, we outline the key revisions made to the manuscript and figures, along with our detailed responses to the editors and referees' comments:

Editor:

A few editorial requests will also need to be addressed:

- Please correct the conflict of interest subheading to "Disclosure Statement and Competing Interests"
 - The corresponding author needs to have an institutional email address
 - The author credits need to be removed from the ms file. All credits are entered during online ms submission.
 - Materials and methods should be called "Methods"
- *Figure Legends - Comments*
- Please note that the legend for figure 4k is missing in the manuscript. This needs to be rectified.
 - Please note that the exact p values are not provided in the legends of figures 1d-e, g, k-m; 2g-h; 3e, h, j, l, n; 4a-g, i; 5b, d-j, l-s.
 - Please note that in figures 1d-e, g, k-m; 2g-h; 4a-g, i; there is a mismatch between the annotated p values in the figure legend and the annotated p values in the figure file that should be corrected.
 - I would like to suggest some changes to the abstract that needs to be written in present tense. Please let me know whether you agree with these changes and please also modify the abstract in response to referee 3's comments, if necessary:

Transcriptional regulation governs gene expression levels, primarily controlled by "cis-acting DNA elements" and "trans-acting protein factors". However, the conventional view that cis-regulation is solely attributable to DNA elements is challenged in this study. Our research indicates that transposon-derived proteins may retain their original DNA-binding preference and exert cis-regulatory effects on nearby genes on the chromosome, thus denoted as "cis-acting factors". Specifically, we show that the ADF-1L protein, derived from the PIF/harbinger transposon, recruits the histone acetyltransferase KAT2B in a MADF domain-dependent manner, facilitating its own nuclear translocation and binding to and cis-regulating its own and adjacent gene 7SL-23. ADF-1L protein also boosts the host's resistance to pathogens by promoting the expression of immune molecule 7SL RNA. In summary, our findings expand the types of molecules that can exert cis-function in gene regulation and underscore the relevance of transposons-derived sequences in the cellular processes.

Response: Thank you for your careful review of our manuscript and your insightful editorial suggestions. We have thoroughly addressed each of your points and made the necessary revisions. Below is a detailed response to the editorial requests:

1.Conflict of Interest Subheading: The subheading has been corrected to "Disclosure Statement and Competing Interests" as per your request.

2.Corresponding Author's Email: The corresponding author's institutional email address has been provided in the manuscript.

3.Author Credits: All author credits have been removed from the manuscript file. As requested, the credits will be entered during the online submission process.

4.Materials and Methods Section: The section has been updated and renamed to "Methods" to conform with your suggestion.

5.Figure Legends:

The missing legend for Figure 4k has been added.

We have corrected the figure legends for figures 1d-e, g, k-m; 2g-h; 3e, h, j, l, n; 4a-g, i; and 5b, d-j, l-s by including the exact p-values.

Mismatched p-values between the figure legends and figure files in figures 1d-e, g,

k-m; 2g-h; 4a-g, i have been rectified.

6. **Abstract:** Regarding your suggestion for the abstract to be written in the present tense, we have agreed with the proposed changes.

We believe that these revisions significantly enhance the manuscript, and we hope it now meets with your approval. Thank you again for your time and effort in reviewing our work.

Referee #3:

Question 1: In their response, the authors state that 'active 7SL genes exhibited different responses to pathogen infections'. These data should be presented in Figure 1, in order to assess/compare to the response of 7SL-23, as well as to CHIP data shown in Figure 2G. Varied responses among more distant copies weakens, rather than strengthens, the argument that 7SL-23 regulation by ADF-1L is based on its physical proximity, a major claim of manuscript (e.g. 'cis-acting' nature highlighted in manuscript):

Response: Thank you for your thoughtful and constructive feedback. In response to your suggestion, we have moved the expression peaks of different active 7SL genes from Fig. EV1 to Fig. 1C. This modification allows for a more direct comparison of the 7SL response to pathogen infections and aligns it with the ChIP data shown in Fig. 2G. As illustrated in Fig. 1C, 7SL-23 exhibits the highest sequencing depth, indicating its highest transcriptional activity, which is further confirmed by Fig. 2G. We believe this heightened activity may be linked to its proximity to ADF-1L.

Moreover, we do not believe that the varied responses of different 7SL copies to pathogens weaken our proposed cis-regulatory model. For instance, while 7SL-16, another active gene, is significantly induced by pathogen infection like 7SL-23, it is not regulated by ADF-1L (Fig. 2G and 2H). This suggests that, while proximity to ADF-1L may influence regulation, other factors also contribute to the transcriptional response. Furthermore, it is noteworthy that all eight 7SL genes, ranging from 7SL-9 to 7SL-16, are all located in adjacent positions on chromosome 12 and show sensitivity to pathogen infection (Fig. 1B and 1C), highlighting a potential correlation

between genomic location and the transcriptional activity of 7SL genes. We have incorporated this discussion into the revised manuscript to further support our proposed model (Lines 140-144 and Lines 206-208). Thank you again for your valuable suggestion.

Question 2: In their response, the authors state that, 'Results showed that ADF-1L knockdown specifically reduces the expression of 7SL-23 without affecting 7SL-16 or 7SL-18, highlighting the selective regulation of 7SL-23 by ADF-1L (Fig. 2G and 2H)'. Rather than infer from CHIP data, the authors should measure the effect of ADF-1L KD on the expression of active 7SL genes (e.g. 7SL-16 and 7SL-18) to show that ADF-1L activity is specific to 7SL-23, and, as authors claim in response, not 7SL-16 or 7SL-18.

Response: Thank you for your valuable feedback regarding the presentation of our results. We would like to clarify that our original intention was to investigate the regulation of ADF-1L on the transcriptional activity of 7SL, rather than its effect on the expression of 7SL. This distinction is important, as the expression of 7SL could still be influenced by other post-transcriptional modifications. Thank you for your reminder, we have accordingly revised "expression" to "transcriptional activity". (Line 204).

In addition, as stated in our previous responses, due to the high sequence conservation between different copies of 7SL (as evidenced by sequence alignment in Fig. EV1A), the designed RT primers may have off target effects, making it impossible to directly measure the effect of ADF-1L knockdown on RNA expression of 7SL-16 and 7SL-18 by designing RT primers. 7SL is transcribed by RNA polymerase III, and the transcription factor complex TFIIC is essential in this process. Therefore, we conducted CHIP experiments to examine the binding affinity between TFIIC and the various 7SL genes to evaluate their transcriptional activity. Our CHIP data clearly demonstrate that ADF-1L knockdown selectively reduces the transcriptional activity of 7SL-23, without affecting the transcription of 7SL-16 or 7SL-18, confirming the specific regulation of 7SL-23 by ADF-1L. To provide

additional clarity on this methodology and to better explain why ChIP experiments were employed to study ADF-1L's regulation, we have included this explanation in the revised manuscript (Lines 196-201).

Question 3: Overall, the inclusion of additional data has strengthened the authors' claims. That said, it remains unclear whether the physical proximity is mechanistically-related to the observed regulation of 7SL-23 by ADF1L. Ectopic expression of ADF-1L also leads to the upregulation of 7SL-23. Given that the 'cis-acting' nature of ADF1-L is central to the manuscript's narrative, more clarification of the novel nature would be warranted. For example, are all transcription factors that regulate their own expression by binding to their own promoters considered 'cis-acting factors'?

Response: The physical proximity between ADF-1L and 7SL-23 on the chromosome plays a crucial role in ADF-1L's regulation of 7SL-23 transcription. First, ADF-1L can selectively regulate nearby 7SL-23 gene without affecting other 7SL genes, such as 7SL-16 and 7SL-18. Second, ADF-1L protein can regulate the acetylation modification state of its own gene and the 7SL-23 locus, which provides further support for this regulation. In addition, the co-directionality of ADF-1L and 7SL-23 transcription, along with ADF-1L's self-regulation, provides a strong theoretical foundation for ADF-1L's cis-regulatory role on 7SL-23. Finally, the transposon origin of ADF-1L offers evolutionary evidence supporting its cis-regulatory function. As for the ectopic expression of ADF-1L, it does not contradict the concept of cis-regulation that we propose. Both ectopic expression and natural translation of ADF-1L aim to increase the production of ADF-1L protein, which in turn promotes cis-regulation.

In traditional transcriptional regulation systems, cis-regulatory elements are DNA sequences located near the target gene that can regulate its transcription. Therefore, we consider "cis-acting factors" to be proteins that can regulate genes located nearby on the chromosome. A transcription factor that only binds to its own promoter to regulate its own expression would not be considered a "cis-acting factor."

In fact, previous studies have shown that transcription factors like p53 (DOI: 10.1089/dna.1995.14.759) and OCT4/SOX2 (DOI: 10.4161/epi.2.1.4067) can regulate their own transcription through cis-acting DNA elements, forming positive feedback loops. However, we do not consider these transcription factors to be "cis-acting factors," as there is no evidence that they regulate the transcription of genes nearby. In contrast, the "cis-acting factor" we propose, such as ADF-1L, not only regulates its own transcription but also regulates the adjacent 7SL-23 gene. We attribute this to the DNA cis-binding preference of the MYC/sant domain of PIF/harbinger transposons. The p53 and C/EBP transcription factors, however, seem to lack the mechanism to regulate neighboring genes. Their regulation of their own expression appears to be a coincidental positive feedback loop that was retained through evolution due to its selective advantage. Based on your suggestion, we have added this discussion to our revised manuscript to further clarify our proposed "cis-acting factor" model (Lines 94-99).

Prof. Tianjun Xu
Shanghai Ocean University
Shanghai 316022
China

Dear Prof. Xu,

I am very pleased to accept your manuscript for publication in the next available issue of EMBO reports. Thank you for your contribution to our journal.

Yours sincerely,
